# Hybrid Satellite–Terrestrial Networks toward 6G: Key Technologies and Open Issues

**DOI:** 10.3390/s22218544

**Published:** 2022-11-06

**Authors:** Syed Bilal Raza Tirmizi, Yunfei Chen, Subhash Lakshminarayana, Wei Feng, Aziz A. Khuwaja

**Affiliations:** 1School of Engineering, University of Warwick, Coventry CV4 7AL, UK; 2Department of Electrical Engineering, Sukkur IBA University, Sukkur 65200, Pakistan; 3Department of Electronic Engineering, Tsinghua University, Beijing 100084, China

**Keywords:** cooperative HSTN, cognitive HSTN, performance evaluation

## Abstract

Future wireless networks will be required to provide more wireless services at higher data rates and with global coverage. However, existing homogeneous wireless networks, such as cellular and satellite networks, may not be able to meet such requirements individually, especially in remote terrain, including seas and mountains. One possible solution is to use diversified wireless networks that can exploit the inter-connectivity between satellites, aerial base stations (BSs), and terrestrial BSs over inter-connected space, ground, and aerial networks. Hence, enabling wireless communication in one integrated network has attracted both the industry and the research fraternities. In this work, we provide a comprehensive survey of the most recent work on hybrid satellite–terrestrial networks (HSTNs), focusing on system architecture, performance analysis, design optimization, and secure communication schemes for different cooperative and cognitive HSTN network architectures. Different key technologies are compared. Based on this comparison, several open issues for future research are discussed.

## 1. Introduction

It has been predicted that smartphones will consume more than 226 EB of data on average per month by 2026 to occupy about 91% of total mobile traffic [1]. Since wireless communication has been so successful, terrestrial cellular networks are now widely used and serve billions of people globally. The transition from the first generation (1G) to the fifth generation (5G) in terrestrial cellular networks has occurred quickly over the last two decades. The evolution of terrestrial networks toward the sixth generation (6G), which is intended to offer higher data rates, lower latency, greater connection, and improved quality of service (QoS), is now taking place in order to fulfill the increasing demand for mobile data on a global scale. Although terrestrial communications have seen rapid advances in both data rates and infrastructure, such explosive growth in the number of users and the supported services requires more resources. To support modern-day data requirements, new technologies, infrastructures, and standards have been proposed [2].

Current terrestrial systems may not be sufficient to provide services that require higher data rates with reliability and up-to-date QoS standards, especially in the harsh oceanic geographic regions [3] critical for maritime communication networks (MCNs), as they suffer from coverage and capacity issues. Additionally, owing to economic and geographic constraints, terrestrial networks are mainly deployed in developed areas, such as urban areas. There are still large numbers of people and devices that remain unconnected even after construction of the 5G network. With the deployment of the 5G network, research on the 6G wireless network comes into focus to overcome these unsolved challenges. The International Telecommunication Union (ITU) has built the Network 2030 group for developing the next-generation wireless network. China has established a project to study the 6G wireless network for 2030 and beyond [4]. In the development targets of the 6G network, peak data rate is expected to reach 100 Gb/s to 1 Tb/s, which is 10 to 100 times higher than that of the 5G network. Latency is expected to decrease to 0.1 ms, which is a tenth of that of the 5G network. Additionally, other targets include higher positioning accuracy, higher energy efficiency, extreme reliability, larger connectivity density, and longer battery life. In the 6G white paper, it has been proposed that the future wireless network must be able to seamlessly interface with terrestrial and satellite networks [5]. Non-terrestrial networks and hybrid networks will play an important role to achieve this.

The use of satellite communication systems overcomes the limitations of terrestrial networks due to their coverage over larger areas, especially in tough terrain, and their ability to multicast and broadcast [6]. However, satellite communication systems have their own challenges and limitations due to their complex deployment, expensive infrastructure, and deep fading at higher frequencies. Due to the high costs of terrestrial networks, satellite wireless backhaul is being considered for rural and remote areas in 6G. Moreover, different satellites have different features, as follows.

Geostationary (GEO) satellites operate in a fixed orbit of 36,000 km. Due to their high altitude, GEO satellites can offer the most extensive coverage. GEO satellites do not require frequent handovers due to their high altitude. However, high altitude also causes propagation delay and signal attenuation.Medium Earth orbit (MEO) satellites are typically used in orbits ranging from 2000 to 36,000 km. MEO satellites have smaller coverage than GEO satellites but much less propagation delay and signal attenuation. Additionally, as the altitude of MEO satellites is variable, MEO orbit resources are generally adequate.The typical operating orbit for low Earth orbit (LEO) satellites ranges from 500 to 2000 km. Because LEO satellites are so much lower in the atmosphere than GEO and MEO satellites, their propagation delay and signal attenuation are substantially less. However their structures become more complex, as more LEO satellites are needed to cover a large area. LEO orbit resources are also comparatively sufficient because the height of LEO satellites is not fixed.Enabling the path to 5G and beyond, the concept of very low Earth orbit (VLEO) has also been explored due to its reduced communication latency. The distinction between VLEO and LEO is the altitude of the former is below 300 km. Operating a telecommunications satellite in this region has many benefits. The closer the satellite is to the Earth, the better the link budget and the lower the latency and the required transmit power. Other benefits include improved frequency reuse capability, a lower radiation environment, and a wider available band, leading to lower rainfall interference and improved launcher uplift capability [7,8].

A satellite network normally consists of a transmitting Earth station and a hub that processes and amplifies the received signal before relaying it back to one or more ground stations for reception. These satellite networks are primarily configured in four basic topologies [9].

Point-to-Point Network: A point-to-point satellite network involves a dedicated link between two sites that are located within the same satellite footprint. This type of network easily supports voice, video, and data transmission.Star Network: The star network is the most commonly used topology for both unidirectional and bidirectional networks as it provides much greater flexibility. This network allows transmission of information in both directions, but data cannot be transmitted directly from one very-small-aperture terminal (VSAT) to another. All information is routed through the hub station.Mesh Network: A mesh network is mainly used for real-time telephony or video-teleconferencing applications. It allows several remote sites to communicate with each other via a single link through the satellite resulting, in a minimal time delay between signal transmission and reception due to its “single hop” nature.Hybrid Network: A hybrid network is basically a combination of both star and mesh topologies. This topology allows the hub to send information to the remotes, with the remotes then able to communicate with other VSAT locations.

In the past decades, several satellite systems were constructed and applied for global communications, such as Globalstar [10], Iridium [11], O3b (other 3 billion) [12], and the Transformational Satellite Communications System (TSAT) [13].

Globalstar [10] is a constellation of 48 LEO satellites to provide low cost satellite cellular communications. The system mainly covers the area between 70 north latitude and 70 south latitude. Each service area is always covered by two to four satellites, and users can access the system at any time.Iridium [11] is a personal communication network of 66 LEO satellites for voice and data services. Each Iridium satellite is equipped with multi-beam antennas and can support on-board processing, switching, and routing. Additionally, each satellite is equipped with four inter-satellite links (ISLs), which guarantees that the entire process of satellite communication can be achieved by one gateway.The ‘other 3 billion’ [12] is a constellation of MEO satellites to help 3 billion people in Africa, Asia, and South America without internet access.TSAT [13] was designed by the National Aeronautics and Space Administration (NASA), the US intelligence community, and the US Department of Defense (DoD). It is a network of GEO satellites for military applications that allows real-time satellite imagery.

The idea of combining satellite and terrestrial communication networks has been proposed, taking into account the fundamental characteristics of both systems. While both satellite and terrestrial networks can deliver high data rates, satellites can expand coverage over broad areas, including rural, urban, and ocean environments. Combining the advantages of both networks, an integrated satellite–terrestrial network architecture is promising for the 6G network to enable global coverage and Internet access and to provide ubiquitous communication support for the Internet of Things (IoT). HSTNs have a number of advantages, including, but not limited to:Diversity of service provision, such as multicast and broadcast;Enhanced coverage with the best possible connectivity and cost;Resource optimization through the best available network selection;Availability of services for critical and emerging communications, such as Internet of Things (IoT), maritime, aeronautical, and railways;Improved efficiency of blockchain by addressing the scalability problem caused by the limited increase in block size and effective neighbors due to fixed bandwidth at each node [14].

Due to the continued interest, more and more organizations, both in industry and academia, are developing ways of utilizing modern infrastructure and technology. Organizations such as Global Information Grid (GIG) [15,16], OneWeb [17], etc., are investing in projects based on modern technologies due to their capabilities and usability in many practical fields, such as Earth observation, geo mapping, disaster response [18], and so on.

Starlink is an LEO satellite network proposed by SpaceX [19]. The Starlink network is designed to provide global high-speed Internet access by using nearly 42000 LEO satellites on different orbits from 340 km to 1150 km. Each satellite can support a transmission capacity of 20 Gb/s using the Ka-band, Ku-band, and V-band. On 5 October 2022, SpaceX launched another 52 Starlink satellites to low earth orbit to provide communication service for 90,000 users in dozens of countries.The OneWeb network consists of 648 LEO satellites in 18 orbital planes at an altitude of 1200 km. The transmission capacity of each satellite is up to 7.5 Gb/s using the Ka-band and Ku-band. The OneWeb network adopts the simple design of transparent forwarding to directly provide users with Internet services through ground gateways. OneWeb has partnered with NewSpace India Limited (NSIL), the commercial arm of India’s national space agency—the Indian Space Research Organization (ISRO)—for its next launch. Launch 14 will fly 36 satellites into low Earth orbit, increasing the OneWeb fleet to 464. Further launches in 2022 and 2023 will complete the constellation [20].Eutelsat Konnect is a very-high-throughput satellite (VHTS) communication satellite built by Thales Alenia Space, the joint venture between Thales (67%) and Leonardo (33%). It was successfully launched by an Ariane 5 rocket from the Guiana Space Center—Europe’s Spaceport in Kourou, French Guiana. Eutelsat Konnect satellite will deliver high-speed broadband and mobile connectivity across Europe, North Africa, and the Middle East. With a Ka-band capacity of 500 Gbps, it has the largest bandwidth and the highest capacity of any geostationary communications satellite ever built in Europe. The satellite uses the most powerful on-board digital processor ever put in orbit, offering capacity allocation flexibility and optimum spectrum use [21].

The HSTN network architecture seems promising for the 5G network and beyond to offer global coverage and Internet access by combining the attributes of both networks and providing ubiquitous communication support for IoT. HSTN consists of the conventional terrestrial networks with satellites for extension of coverage, as shown in Figure 1. The terrestrial network, which uses cellular networks and backhaul links to connect BSs to the core network, offers broadband services for developed areas. Satellites’ extensive coverage allows provide connectivity to everyone and everything in rural and remote places. Users who are not covered by terrestrial networks can use their own terminals to connect to the satellite network, albeit this may have capacity restrictions depending on the users and terminals. Additionally, satellite-based BSs and access points can be installed to deliver relatively broadband service, and users can connect to these devices using 6G or Wi-Fi technologies.

A large number of European Union (EU)-funded projects attempt to expand the scope of global connectivity using cutting-edge concepts and technologies. The following list includes some of the EU-funded initiatives under Horizon 2020 to establish connectivity in “problem terrain” away from dense urban and suburban populations.

The shared-access terrestrial–satellite backhaul network enabled by smart antennas (SANSA) is a hybrid terrestrial–satellite backhaul network that is built on three key concepts: (i) seamless integration of the satellite segment into terrestrial backhaul networks, (ii) a terrestrial wireless network capable of changing its topology in response to traffic demands, and (iii) spectrum sharing between the satellite and terrestrial segments. Combining these two technologies will result in a flexible solution that can efficiently route mobile traffic while providing resilience against link failures or congestion and enabling quick deployment in remote areas [22].Satellite and terrestrial network for 5G (SAT5G) aims to provide affordable “plug-and-play” satellite communication (satcom) solutions for 5G to enable telecom operators and service providers to speed up 5G deployment in all geographies while offering new and expanding market opportunities for satcom industry players. The overall SAT5G objective is to deliver ubiquitous end users with at least 50 Mbps 5G broadband service and optimize satcom within the 5G network infrastructure to provide services in the following verticals: Media and Entertainment, Transportation, Health, Logistics, and Agriculture Industries in developed and emerging markets [23].The 5G agile and flexible integration of satellite and cellular (5G ALL-STAR) project expands on the success and collaboration experience of 5GCHAMPION [24] to design, develop, evaluate, and trial multi-connectivity based on multiple access, combining cellular and satellite access technologies to support seamless, dependable, and all-encompassing broadband services. In order to provide broadband and low-latency 5G services, 5G ALL-STAR will create (i) a 5G cellular millimeter wave (mmWave) access system, (ii) a new radio-based satellite access system that is feasible, (iii) multi-connectivity support based on cellular and satellite access, and (iv) spectrum sharing between cellular and satellite components. The integration framework will be emphasized by the 5G ALL-STAR project, with the goal of integrating several potential access networks (such as satellite access and cellular access based on 4G or 5G) under a single core network for 5G [25].Hybrid networks for global internet access is a worldwide viewpoint to integrate the best of satellite and terrestrial communications into hybrid networks and to connect the world to the internet. With the goal of connecting an additional 1 million users to the internet and generating significant economic and societal value in underserved markets of emerging economies in the developing world and rural and remote areas in the developed world, this project is the first step toward the successful market introduction of hybrid communications solutions [26].

Moreover, the European Cooperation in Science and Technology (COST) is a funding organization for the creation of research networks, known as COST actions. These networks offer an open space for collaboration among scientists across Europe (and beyond) and thereby give impetus to research advancements and innovation, contributing to advance the state-of-the-art in convergence between fixed and mobile terrestrial and satellite telecommunications, Earth observation and navigation systems, global integrated networks (including disaster management and relief), radio wave propagation (including optical free space links), and atmospheric remote sensing techniques and meteorology [27].

A comparison of different satellite and terrestrial networks with respect to speed and latency is shown in Table 1.

In this work, we aim to provide a comprehensive survey of cooperative and cognitive HSTNs. Our motivation is driven by the potential of HSTNs as future wireless communication networks for providing broader coverage and higher data rates with reliable and efficient quality of service and the requirement for understanding their performances for optimal deployment. Several surveys have been published so far, covering various aspects of satellite, satellite–terrestrial, and satellite–air–terrestrial integrated communication networks. For example, Chowdhury et al. in 2006 published a survey of handover schemes in satellite communications [28]. Chini et al. reviewed network architectures, operational systems, services, and research issues in mobile satellite systems in 2011 [29]. Research conducted by the small satellite community, including design parameters for inter-satellite communications and solutions that enable operations in small satellite systems, has focused on inter-satellite communications for small satellites [30]. Niephaus et al. provided state-of-the-art satellite and terrestrial network convergence and focused on functionality to optimize traffic distribution, architectures, and related adaptations to provide support for converged satellite–terrestrial networks [31]. Challenges related to the performance of optical communications in integrated satellite–terrestrial networks and techniques for the mitigation of atmospheric side effects were reviewed by Kaushal et al. in [32]. Liu et al. reviewed space–air–ground integrated networks, focusing on network design, resource allocation, mobility management, open challenges, and future directions [33].

P. Wang et al. discussed the convergence of satellite and terrestrial networks with a prime focus on recent developments in satellite terrestrial networks, system architectures, performance evaluation and simulation platforms [34]. Rinaldi et al. in 2020 reviewed non-terrestrial network (NTN) use cases and architectures, a satellite network roadmap and the role of NTN in cellular communications, and NTN open issues and research directions beyond 5G [35]. A survey of the state-of-the-art technologies for models X, L, and Y of cooperative HSTNs was provided in [36]. Future research directions toward establishing a cell-free, hierarchical, decoupled HSTN were presented, and open issues to envision an agile, smart, and secure HSTN were outlined. In [37], the authors reviewed state-of-the-art hybrid satellite–terrestrial MCNs for future maritime communications and proposed an environment-aware, service-driven, and integrated satellite–air–ground MCN that included external auxiliary information such as sea level and atmospheric conditions. The list of published surveys of HSTNs and their contributions are summarized in Table 2.

Although all the surveys mentioned above have provided deep insights and diverse perspectives about satellite, satellite–terrestrial and satellite–air–terrestrial integrated networks, there is still the need for a comprehensive review of HSTNs, especially their integration architectures, such as cooperation and cognition; key technologies, such as non-orthogonal multiple access (NOMA), beamforming, software defined radio; and the state-of-the-art in physical-layer security of HSTNs.

Our work will mainly focus on providing a review of the up-to-date work on HSTN integration for 5G and beyond and key technologies with respect to performance evaluation studies using key measures such as outage probability (OP), symbol-error probability (SEP), symbol error rate (SER), average symbol error ratio (ASER), ergodic capacity (EC), signal-to-interference-plus-noise ratio (SINR), and bit error rate (BER) for different cooperative and cognitive technologies used in HSTNs.

The rest of this paper is structured as follows. Section 2 provides a comprehensive and up-to-date overview of HSTN cooperative networks. Section 3 provides an overview of cognitive HSTNs, including spectrum sharing, performance parameters, hardware impairment, and interference. Section 4 discusses various potential applications of HSTNs. Section 5 discusses key technologies in cooperative and cognitive HSTNs. Section 6 discusses open issues, and Section 7 provides a conclusion.

## 2. Cooperative HSTN

### 2.1. Coordinated HSTN

While the terrestrial network has been effectively implemented in developed areas to offer broadband Internet access at low cost, the satellite network can offer ubiquity with wide coverage. Thus, these two systems can cooperate for mutual benefits. Figure 2 shows the structure of coordinated HSTN. Coordinated HSTN is a promising architecture because it makes use of the strengths of both networks instead of choosing only one of them. Using the same physical-layer protocols, unified terminals may effortlessly access either the satellite network or the terrestrial cellular network. Higher network efficiency and user experience can be achieved when integration advances from the top level to the bottom level, at the expense of higher implementation complexity and deployment costs.

Broadband services can be supported via the terrestrial network at fairly low cost. However, due to financial and geographical limitations, the coverage of each BS is quite narrow, which restricts the expansion of terrestrial networks. On the other hand, because of its top-down design, the satellite network can offer extensive coverage. However, the satellite network generally experiences large signal attenuation and latency. To minimize the inherent communication delays, path loss, and deployment costs of the satellite network [38], aerial networks have been suggested as a mitigating part of HSTNs. In addition to their versatility and low cost, these elements have lower latency and can operate with spot beams delivering more capacity to ground users and a greater diversity of options.

Due to their inexpensive cost and tremendous versatility, unmanned aerial vehicles (UAVs) have received much interest in recent years for use as aerial BSs. In contrast to cellular communications, UAVs can freely and adapt their movement in a 3D space, making it possible to establish line-of-sight (LOS) communication links and to prevent signal blockage and shadowing. UAVs are also seen as a possible method of achieving ubiquitous connectivity for IoT, which is fascinating from a technological perspective. However, aerial networks, including UAVs, incur significant energy consumption due to propulsion and hovering. Their intrinsic mobility could impair network performance without proper coordination and countermeasures [39]. Figure 3 shows the structure of an HSTN assisted by UAVs.

To relay signals to the users, networks use different relaying schemes such as amplify-and-forward (AF), fixed decode-and-forward (FDF), and selective decode-and-forward (SDF) [40,41]. AF works by amplifying both the source signal and noise and forwarding them to the users, which may lead to performance degradation. In FDF transmission, source messages are first demodulated and then remodulated before being forwarded to users, while in SDF, only relays that can correctly demodulate the source message are allowed to retransmit the signal. This prevents the relay network from transmitting erroneous messages to the destination. Although communication noise can be reduced in the DF protocol, this is of higher complexity, and the relay process is limited by the satellite–relay channel. If the relay cannot decode the source signal, it will make no contribution to the network.

The utilization of UAVs in HSTNs has been expanded to include maritime coverage enhancement through UAVs integrated MCNs. A hybrid satellite–UAV–terrestrial network solution was provided in [42]. Opportunities and challenges for integrating UAVs into existing MCNs for enabling 5G on the ocean were also investigated. For higher benefit-to-cost ratio, orchestration of hierarchical links for a hybrid satellite–UAV–terrestrial MCN for 5G coverage enhancement was discussed in [43]. Similarly, coordination of UAVs with existing MCNs for on-demand communication coverage enhancement was discussed in [44].

Multicast communication with rate-splitting multiple access (RSMA) was studied by the authors of [45] for a satellite–aerial integrated network where both satellite and UAV components are controlled by a network management center and operate in the same frequency band. The intended performances of interference suppression, spectral efficiency, and hardware complexity were obtained by taking into account a content delivery scenario where the UAV subnetwork adopts the RSMA to support enormous access by IoT devices.

In [46], a cooperative satellite–aerial–terrestrial system including a satellite transmitter (S), a group of terrestrial receivers (D), and an aerial relay (R) was considered. Specifically, considering the randomness of S and D and employing stochastic geometry, the coverage probability of R–D links in non-interference and interference scenarios was studied, and the outage performance of S–R links was investigated by deriving an approximate expression for the outage probability. Moreover, an optimization problem in terms of the transmit power and the transmission time over S–R and R–D links was formulated and solved to obtain the optimal end-to-end energy efficiency of the considered system.

In [47], a space–air–ground integrated network (SAGIN) from the perspective of cooperative communications was reviewed. New relay networking technologies were introduced, and channel models were reconstructed by considering high-altitude platform (HAP) mobility and a practical propagation environment.

The ASER performance of an AF-based HSTN based on distributed space–time coding (DSTC) was studied in [48]. Shadowed Rician and Rayleigh fading models were considered for the satellite–relay, satellite–destination, and relay–destination links. The study found that a DSTC-based HSTN performs better than a traditional HSTN for the given fading models. Further, the authors of [49] theoretically analyzed the performance of a distributed space–time-coded HSTN, where the relay and the satellite cooperatively transmit Alamouti-coded signals to the destination.

Due to the large difference between the satellite and terrestrial links, channel estimation in HSTNs is not straightforward. Due to unavailability of channel state information (CSI), the authors of [50] proposed a joint cooperative link estimation and symbol detection scheme. ASER, average capacity, and diversity order of an AF-based HSTN with estimated CSI were derived. The authors concluded that the quality of the terrestrial link channel estimate does not affect the analytical results significantly. Thus, system performance was found to largely depend on the quality of the cooperative link channel.

For increased bandwidth demands, 5G backhaul networks operate in millimeter wave frequencies. In such a scenario, rain attenuation becomes a significant fading mechanism. The authors of [51] evaluated the end-to-end performance of an AF-based cooperative-diversity HSTN for backhauling applications operating at millimeter frequencies. An elegant stochastic channel model was employed in order to generate a rain attenuation time-series spatially correlated for the multiple links of the system. However, cooperative strategies in the relay and various reception techniques were not discussed.

In [52], the exact symbol-error probability and diversity analysis of a hybrid satellite–terrestrial cooperative network was investigated, focusing on the mobile relays over independent but not identically distributed fading channels. Mobile relays can be both moving vehicles and handheld cell phones. A selective DF scheme was implemented to avoid relaying of erroneous signals to the users. A moment-generating function (MGF) approach was used get the closed-form expressions for SEP. The authors also provided evidence of an achievable diversity order of *N* + 1 for *N* number of relays.

The work in [53] presented the performance analysis for a selective DF cooperative HSTN over time-selective fading links with multiple-input–multiple-output (MIMO) and space–time block coding for multiple relays. The satellite–relay links experienced non-identical time-selective shadowed Rician fading, and the relay–destination terrestrial links were assumed to be non-identical time-selective generalized Nakagami faded. Closed-form expressions were derived for the per-frame ASER and the asymptotic SER floor. A list of major contributions to HSTN coordinated networks is provided in Table 3.

### 2.2. HSTN Relay Network

To enhance the performance of HSTNs, satellite signals may be relayed independently towards users [54]. Thus, the users of an HSTN can receive signals from both the satellites and the relays to take advantage of this spatial diversity from hybrid systems and to overcome the masking effect in satellite networks. This leads to a HSTN relay network, as shown in Figure 4. Many HSTN relay networks have been proposed in recent years using AF, FDF, and SDF for outage, error, and ergodic capacity analysis in the presence of co-channel interference (CCI), multi-antenna, mobile nodes, shadowing, CSI, and other conditions.

#### 2.2.1. Single Relay Node

Due to its many application scenarios, such as a communication service for a subscriber who is walking into a home or shopping mall and then staying within the building, personal wireless demands have drawn significant attention in response to the explosive growth of mobile access demands. Accordingly, the transmission connections of HSTN relay networks may significantly worsen in urban settings because of the channels’ substantial penetration loss, particularly in high frequency ranges. Reconfigurable intelligent surfaces (RISs), a unique idea with the goal of intelligently regulating the wireless environment to improve the quality of service, have been proposed as a possible solution to this problem [55].

Similar to [55], the unique concept of utilizing refractive RISs was developed in [56] to deal with the issue of blocked satellite communications to enhance the QoS of HSTN relay networks by minimizing the total transmit power of the satellite and base station while satisfying rate requirements of multiple users.

Unlike [52], which did not consider co-channel interference (CCI), the authors of [57] investigated the effect of CCI on an HSTN, and for the first time, MGF-based analytical expressions of the ASER were derived. The considered system has a terrestrial relay with DF, and each node has only one antenna. The satellite–user and satellite–relay links undergo shadowed Rician fading, whereas the relay–destination link follows Rayleigh fading. The investigation found no apparent effect of CCI on diversity order. However, the ASER degraded because of reduced array gain of the system due to CCI.

The effect of multiple interfering sources at relay nodes was investigated in [58] when a direct link was not available due to blockage from large obstacles (shadowing), severe attenuation (path loss), or multi-path fading. The ASER performance of an AF-based HSTN was evaluated when the satellite–relay link was assumed to follow Rice fading and the relay–destination and interferer–destination links followed Nakagami-m fading. The authors found that, depending on the channel, interference, and network parameters, the overall performance of the system can be severely degraded by multiple-CCI at the relay.

In [59], ergodic capacity, OP, ASER, and the impact of key performance parameters, such as the number of antennas, channel coefficients, and CCI power, were studied for the first time in the case where both the satellite and the destination have multiple antennas in an AF-based HSTN with CCI at both the relay and destination nodes. Similarly, CCI was found to significantly degrade system performance, although the maximal diversity order was still achievable.

Extending [57] to a more general case, the authors of [60] considered the performance of an HSTN when both relay and destination nodes were affected by CCI and noise. System performance was analyzed for an AF protocol, and the BER performance was investigated in the case of multiple relays. System performance was better when the best relay selection scheme was used instead of using all the relays.

The impact of outdated CSI and CCI at the relay on a multi-user single-antenna AF-based HSTN with shadowed Rician fading at satellite links and Nakagami-m fading for terrestrial links was studied in [61]. The performance of the considered system was found to be critically affected due to CCI and outdated CSI.

In [62], the authors conducted a comprehensive performance analysis of a multi-user AF-based HSTN by deploying multiple antennas at the satellite and users with shadowed Rician fading for satellite links and Nakagami-m fading for terrestrial links. Opportunistic scheduling of users with outdated CSI and AF relaying in the presence of CCI signals was considered. The authors substantiated that the system diversity order is greatly influenced by the number of antennas, users, and levels of CSI and CCI but independent of the correlation and fading parameters of the LOS in multi-antenna satellite links.

By employing satellites equipped with multiple antennas, the impacts of antenna correlation, CCI, and outdated CSI on the ergodic capacity of an HSTN [61] was considered for both correlated and uncorrelated shadowed Rician fading at the satellite links in [63]. Better system performance was observed by increasing the number of antennas at the satellite.

Extending their work from [61,63], the average symbol-error probability (ASEP) performance of an AF relay-based multi-user HSTN with multiple antennas at the satellite was evaluated in detail in [64]. By considering CCI at the relay and an outdated CSI for terrestrial links with shadowed Rician and Nakagami-m fading for satellite and terrestrial links, respectively, asymptotic SEP expressions at high SNRs were derived to gain insights into the system’s error performance.

In the case of multiple users, the OP was studied for AF-based HSTN in [65] for frequent heavy shadowing (FHS) and average shadowing (AS) scenarios. The authors found that AS outperforms FHS and that system performance improves with an increased number of user nodes due to more coding gain. Similarly, assuming an opportunistic selection DF relay scheme, OP of the integrated satellite–terrestrial network with multiple antennas at the relay was studied in [66] using maximal ratio combining (MRC) at the destination.

Suggesting superiority of multi-antenna relays, the performance of an HSTN in the case of correlated Rayleigh fading at the terrestrial channel with multi-antenna relays was investigated in [67]. Asymptotic OP expressions at high SNRs were developed to find the diversity order and the impact of the number of antennas and users and the satellite transmit power on the system performance. Considering maximum ratio transmission (MRT) and MRC at the relay and selection-combining at the destination, OP was analyzed.

For a multi-user DF integrated satellite terrestrial network with a multi-antenna relay for which the satellite–relay link undergoes shadowed Rician fading, while the relay–user links experience correlated Rayleigh fading, a new probability density function (PDF) to statistically characterize the square sum of independent and identically distributed (i.i.d.) random variables was proposed in [68].

In [69], the performance of the two-way AF-based HSTN was evaluated, with the satellite–relay links modeled as k−u shadowed fading for more accuracy but with a more complex mathematical model, and the user–relay links were modeled as Nakagami-m fading. The exact OP, ASEP, and achievable rate expressions were derived and provide insights into the impact of fading channels on the considered system.

Proposing a new method to calculate the instantaneous rain attenuation rather than a value averaged over the entire rainfall event, in [70], the performance of an AF-based mmWave HSTN was evaluated considering the effects of both rain attenuation and the unavailability of LOS links due to masking. Joining the effects of a dynamic rain attenuation model with a statistical channel model for describing mmWave satellite and terrestrial links in a new channel model, the influence of rain attenuation on mmWave HSTN system design was found.

Motivated by the lack of work discussing the effect of hardware impairment in HSTNs, ref. [71] investigated the performance of a two-way HSTN with hardware impairment. Outage analysis, signal-to-noise-and-distortion ratio (SNDR), and throughput of the single-antenna satellite, terrestrial, and relay node system were analyzed. The work suggested that smaller impairment of the terrestrial relay would lead to better system performance.

An in-depth performance evaluation of a power-splitting NOMA-based multi-antenna HSTN with energy harvesting relays was provided and the closed-form expressions of outage probabilities and ergodic capacities for both AF and DF relaying protocols were derived to perform a comprehensive analysis in [72]. Considering the Rice and Nakagami-m fading distributions for the satellite and terrestrial channels, respectively, the authors of [73] evaluated the ergodic channel capacity of a DF-based single-frequency HSTN when the satellite–destination link is not available due to heavy fading.

For a DF-based HSTN where no LOS link is present due to masking, and the satellite and terrestrial links follow shadowed Rician and Rayleigh fading, respectively, the authors of [74] obtained analytical channel capacity for optimal power and rate adaptation and truncated channel inversion adaptive transmission protocols with improved performance at low and medium SNR ranges.

To improve reliability and efficiency, a hybrid automatic repeat request (HARQ) in an AF-based HSTN was studied in [75], where delay-limited throughput for both CSI-assisted and fixed-gain relaying protocols were theoretically analyzed with respect to both chase-combining and incremental-redundancy schemes.

#### 2.2.2. Multiple Relay Nodes

The ergodic capacity and ASER of a fixed-gain AF-based HSTN was considered in [76]. A fixed-gain AF-based HSTN was investigated because of low-complexity system design. The effects of relay number, channel parameters, and modulation formats on system performance were also studied and validated through computer simulation. The authors found that the modulation format can change the system’s array gain, and the ASER performance of binary-phase shift keying (BPSK) was found better than that of eight-phase shift keying (8PSK). Similarly, the ASER of an AF-based HSTN was investigated in [77] under the influence of CCI due to frequency reuse. The degradation of ASER performance was observed due to shadowing in satellite links causing reduced array gain.

Extending the work in [65], the authors of [78] considered the performance of a multi-relay and multi-user HSTN. The authors proposed a user–relay selection scheme and derived the exact and asymptotic OP expressions for both fixed- and variable-gain relaying.

A performance evaluation of a DF-based HSTN over time-selective fading links due to mobile terrestrial nodes with multiple relays was considered in [79]. The satellite–terrestrial links were assumed to be non-identical time-selective shadowed Rician fading whose parameters depended on the satellite elevation angle. The relay–destination terrestrial links were assumed to be nonidentical time-selective generalized Nakagami fading. The authors observed that the time-varying nature of the links led to discernible degradation of the end-to-end performance.

By considering a more-generalized HSTN with multiple relays and an available direct satellite–destination link, the authors of [80] addressed asymptotic analysis of the ergodic capacity for both AF and DF protocols based on MGF and proposed a simple relay-selection strategy requiring only statistical CSI with low system overhead.

The authors of [81] investigated the ergodic capacity of the downlink in an HSTN for which fading of the satellite channel followed shadowed Rician distribution and fading of the terrestrial channel followed Rayleigh distribution. The closed-form expressions for the ergodic downlink channel were derived, and the system performances for both AF- and DF-based HSTNs were analyzed. A relay selection strategy with multiple relay nodes was discussed. The authors found that the ergodic capacity of a DF-based HSTN is larger than that of an AF-based HSTN under similar channel conditions. The ergodic capacity decreased with an increasing number of participating relays. However, the best relay selection strategy achieved larger ergodic capacity at the cost of channel estimation at each relay.

For communication systems operating over fading channels, OP is an important performance parameter as it provides insight into the communication system’s design. The authors of [82] evaluated the performance of an HSTN in terms of OP with the best relay selection scheme. The selective DF mode was used for the network, which consisted of a single destination node and multiple relays. For the first time, closed-form expressions of outage probability were derived and validated using Monte Carlo simulation.

In [83], to improve system performance, the authors proposed the incorporation of content and uniform content wireless caching placement schemes for HSTNs, resulting in improved system OP performance. In the case of multi-relay systems under the same asymmetric fading channels, maximizing the satellite’s received power and Nth worst relay selection schemes were also proposed.

A novel partial relay selection scheme (PRS) for a DF-based HSTN that reduces relay selection overhead without compromising system performance was proposed in [84]. Assuming the unavailability of a direct satellite–destination path due to masking, asymptotic OP analysis was conducted in practical environments in which the satellite–relay and the relay–destination links were subject to generalized-k and Nakagami-m fading.

Using optimal relay selection, the authors of [85] evaluated the performance of a single-antenna HSTN. By considering the correlated Rician fading between the relay and terrestrial nodes and assuming availability of the channel state information from the satellite–relay and relay–destination links at the satellite for optimal relay selection, an increased outage performance gap with an increase in the number of cooperative relays was found.

The authors of [86] evaluated the performance of an HSTN with hardware impairment in the case of multiple relays using a switch-and-stay combining scheme; they proposed a new link-selection scheme in the satellite–terrestrial network. For better utilization of the spectrum, two-way relays were found to enhance system performance.

The authors of [87] analyzed the impact of hardware impairments (HIs) on a two-way HSTN considering both AF and DF protocols and an opportunistic relay selection scheme. Two-way terrestrial relays were equipped with multiple antennas. The authors concluded that the level of HI, number of relays, and number of antennas significantly affect system performance.

In [88], a multi-relay selection (MRS) scheme was proposed to enhance the performance of a multi-relay DF-based HSTN for which the satellite, relay, and the terrestrial nodes were each subjected to hardware impairment.

Extending their work in [71], the authors of [89] established a general framework of uplink satellite terrestrial relay networks by considering both the impact of HI and CCI, in which a single-antenna user communicates with a satellite via multiple terrestrial relays with multi-antenna configurations and a partial relay selection scheme. To gain further insights into the joint impact of HI and CCI, asymptotic OP and throughput expressions were also derived at high SNRs.

Similar work was reported in [90] for a multi-relay, multi-user mmWave AF-based HSTN. Based on rain attenuation, a new relay selection scheme was proposed, and closed-form expressions of the OP for both CSI-assisted and fixed-gain relaying were obtained.

Aiming to improve HSTN performance, the authors of [91] utilized the AF relay concept as a virtual MIMO. Independent, non-identical, shadowed Rician faded channels for satellite links and Nakagami faded channels for terrestrial links were considered. The system showed improved performance when the LOS link for the satellite–destination or satellite–relay links were less faded, the number of cooperative relays increased, better transmission protocol was used, the number of users decreased in the code division multiple access (CDMA) system, or CDMA orthogonal codes were used.

#### 2.2.3. Aerial Relays

Aerial relays can provide fast and reliable coverage in cases of natural disasters for which the terrestrial infrastructure is not available or for poor communication quality due to deep shadowing and line-of-sight (LOS) signal loss [34,92]. The utility of aerial segment of HSTNs is not limited to relay networks as it can also be used as a base station or user equipment, especially in emergency situations [34].

Assuming perfect CSI for satellite–relay and relay–destination links and considering the impact of satellite antenna patterns and path loss in the channel models, an analytical framework for a multi-antenna unmanned aerial vehicle (UAV) relay-based HSTN was defined in [93]. Asymptotic OP expressions at high SNRs were derived to gain full insights into system performance when the satellite link was subject to shadowed Rician fading and terrestrial links were subject to Nakagami-m fading. Guidelines for future research were also provided.

Considering an AF-based dual-hop satellite relay system operating over 10 GHz, the authors of [94] obtained OP expressions by considering path loss and satellite patterns and evaluated the performance of the system, as rain attenuation becomes a prime factor affecting the satellite links.

Similarly, ref. [95] investigated the OP of an HSTN comprised of a multi-antenna satellite communicating with a ground receiver via multiple AF 3D mobile UAV relays under opportunistic relay selection, and the performances of the proposed 3D mobile UAV relaying versus fixed-altitude mobile UAV relaying and fixed-distance static relaying schemes were compared.

The authors of [96] considered network traffic as an important reference point for the true reflection of the current load on a network segment. An adaboost-based link planning (ALP) algorithm was proposed for faster relay selection and improved load balancing. For the node performance evaluation, an auto-regressive moving average (ARMA)-based network traffic prediction scheme was also utilized. In [97], the superiority of UAV 3D mobile relays was established over static relays by comparing the OP performances of both systems.

In this section, a detailed overview of cooperative HSTNs was provided. State-of-the-art coordinated and relay-based HSTNs were discussed. Different relay scenarios, such as single and multiple relays and aerial relays, were presented. A list of major contributions to HSTN relay networks is provided in Table 4.

## 3. Cognitive HSTN

To address the limited spectrum availability due to the increased demand for broadband multimedia applications, cognitive radio (CR) has become an active and well-researched topic in the field of wireless communications [98] and been incorporated into HSTNs [99]. In cognitive radio, secondary users are allowed to share the spectrum with a primary user network, provided secondary spectrum access will not deteriorate the QoS of the primary network [100]. By sharing spectrum resource between the two networks, higher spectrum efficiency can be achieved, alleviating the pressure of scarce spectrum resources. However, existing infrastructure needs to be updated to support the different frequency bands in conventional satellite and terrestrial networks. Additionally, the cooperation of satellite and terrestrial operators is required for interference mitigation and more efficient resource management.

In the CR underlay model [101], the transmit power of a secondary user is strictly constrained to satisfy the interference temperature at the primary user. In the overlay model [102,103], the secondary users assist primary transmission through cooperative communication techniques.

In recent years, several works have been presented on spectrum sharing [100,104,105,106,107,108,109,110] and cognitive HSTNs, focusing on underlay [101], overlay [102], broadband cognitive radio schemes [111], energy efficiency [112,113,114], power allocation and control [99,115,116,117,118,119,120,121,122], performance evaluation and optimization [123,124,125,126,127], ergodic capacity [128], outage analysis [129,130,131], the impact of imperfect channel state information (CSI) [132,133], interference [134], and hardware impairment [133,135,136]. Figure 5 shows a cognitive HSTN with cognitive satellite–terrestrial radios (CSTR).

### 3.1. Spectrum and Power Allocation

Since the radio spectrum is a valuable resource, it is essential to use it efficiently. Spectrum sharing has also been incorporated in cognitive HSTNs for the efficient use of the allocated spectrum without compromising the quality of the networks sharing the spectrum.

Focusing on the process-oriented data transmission optimization problem due to uncertain UAV flight, the authors of [123] discussed spectrum sharing between UAVs and satellites in a cognitive satellite–UAV network.

To address the problem of limited information exchange between satellite and terrestrial networks and limited overhead of each network for CSI acquisition, ref. [104] proposed a large-scale, CSI-based robust interference mitigation method for a cognitive HSTN. By adopting random matrix theory and variable substitution, the problem was simplified to a max–min problem. The optimal solution, as a saddle point of the optimized objective function, was found with reduced computational complexity.

An opportunistic secondary network selection (OSNS) scheme that minimizes the outage probability of the primary satellite communication through AF-based relay cooperation was proposed in [100] for a spectrum-sharing HSTN. To examine the achievable diversity gains, asymptotic expressions for both the primary and secondary network were derived.

Similarly, for an overlay spectrum-sharing HSTN wherein a primary satellite network coexists with multiple terrestrial secondary networks, partial secondary network selection (PSNS) and OSNS schemes were proposed to improve the outage performance of primary satellite communication through AF-based relay cooperation [105]. Asymptotic OP expressions for both primary and secondary networks were derived in a high SNR region to quantify the achievable diversity order of the considered system.

For a more general scenario of distributed base stations with multiple antennas, the work in [137] evaluated the performance of a cognitive HSTN for which there was concurrent forward link transmission of a primary multi-beam satellite network and a secondary terrestrial mobile network, and where interference to the primary network was below a limit. Characterizing the interference by using stochastic geometry tools, the performances of these schemes were also analyzed in terms of outage probability for both satellite and terrestrial users along with the spectral efficiency of the secondary system to investigate the impact of interference temperature on the average number of successfully transmitted symbols.

Reference [106] provided an overview of HSTN spectrum-sharing techniques such as spectrum sensing, databases, beamforming, beam hopping, adaptive frequencies, and power allocation, and their applications in different scenarios. The citizens broadband radio service (CBRS) concept in the 3.5 GHz band and its rival the licensed shared access (LSA) system in Europe were also discussed.

For a cognitive network in which a satellite network is the primary and a cellular network is the secondary, ref. [138] proposed a novel cooperative cognitive transmission scheme in which the secondary network helps the communication of the primary network to overcome rank deficiency and transmits and receives its own streams without causing interference to the satellite network beyond a certain threshold.

By utilizing knowledge of location information, channel distributions, and antenna radiation patterns in practical environments, and defining an exclusion zone for the primary satellite user to determine the geographical position inside which the cognitive radio approach is not permitted to operate, ref. [139] investigated statistical modeling in cognitive HSTNs, where the terrestrial system acts as the cognitive network and the satellite system is the primary user. Critical factors in deployment, such as channel fading severity, base station location area, and elevation and azimuth angles of the base station with respect to the satellite user were evaluated.

Exploiting the potential of RF-based energy harvesting (EH), ref. [107] studied an EH-based overlay spectrum-sharing HSTN. The secondary transmitter (ST) was assumed to have RF-based EH capability. Thereby, it can act as an AF relay for primary satellite communications while sharing the spectrum for its own transmission. Based on the overlay mode, the ST splits its harvested power to relay the satellite signal and to transmit its own signal. For further insights into the system performance, both heavy shadowing (HS) and average shadowing (AS) scenarios for the satellite links were considered, and the energy efficiency of the primary satellite network assuming a delay-limited scenario was analyzed.

Similarly, a cognitive AF relay-based HSTN was discussed in [140] with a power-splitting-based EH in which a secondary transmitter with EH ability employs power-splitting relaying protocol.

By using an auction mechanism, the authors of [141] looked into multi-channel cooperative spectrum sharing in hybrid satellite–terrestrial IoT networks. In exchange for spectrum access, selected secondary terrestrial-IoT cluster heads helped the primary satellite users transmit through cooperation. With the right secondary network selection and corresponding power allocation profile, the authors proposed an auction-based optimization problem to maximize the sum transmission rate of all primary satellite–IoT receivers while still meeting the minimum transmission rate for each terrestrial–IoT network.

Power allocation is a technique used to distribute the total available power at the transmitter. In cognitive HSTNs, power allocation plays an effective role in improving efficiency and prolonging the lifetime of the network. It helps efficient transfer of information from the source to the destination.

The authors of [99] proposed the concept of cognitive radio for HSTNs with the help of a hybrid satellite–UWB (ultra-wideband) communication system for personal area networks (PANs) for short-range ground communication and hybrid satellite–WRANs (wireless regional area networks) for long-range ground communication. The concept of 3D spatial reuse of the spectrum, integrated architectures, enabling technologies, key issues, and future challenges of cognitive hybrid satellite terrestrial networks were presented in detail.

In [111], a cognitive radio-based experimental platform using universal software radio peripheral (USRP) and personal computer for an HSTN was demonstrated, and a cooperative spectrum-sensing algorithm for mobile users of the terrestrial and space segments was proposed. A detailed overview of the application scenarios of the system along with a future broadband-cognitive HSTN using three GEO satellites and orthogonal and non-orthogonal multiple access techniques based on compressive sensing and key issues related to the applications of cognitive radio to future broadband communication systems toward 5G were provided.

An efficient power allocation scheme for the terrestrial network considering a novel cognitive HSTN was proposed in [128]. The concept of effective capacity was employed to ensure the QoS provision for the terrestrial network, and a specific outage probability was used to ensure the quality of the satellite link. To compliment the proposed resource management scheme, both terrestrial and satellite system quality requirements were investigated.

A short-term and a long-term optimal power control scheme for an uplink underlay-cognitive HSTN was proposed in [115] with the aim of maximizing delay-limited capacity and outage capacity with different constraints, namely, transmit power limits, interference power constraints, satellite link shadowing conditions, and terrestrial interference link fading severity, to ensure communication quality for the primary terrestrial user.

To maximize the energy efficiency of the cognitive satellite user and ensure the interference at the terrestrial primary user is below a certain threshold, energy-efficient optimal power-allocation schemes for real-time and non-real-time applications in underlay-cognitive HSTNs were proposed in [112] using an average interference power (AIP) constraint for the primary terrestrial user and average transmit power (ATP) and peak transmit power (PTP) constraints for the secondary satellite user.

For optimal power control in cognitive HSTNs, the case of imperfect CSI for both satellite and terrestrial interference links was discussed in [116]. Pilot-based channel estimation and a back-off interference power constraint were adopted for the satellite and terrestrial interference links.

Another power allocation scheme for a cognitive two-path successive-relay (TPSR) HSTN with full-interference cancellation (FIC) operating in AF mode and maximizing system capacity under predetermined co-channel interference to the primary user network was proposed in [142].

Taking a more general approach from the models presented in [115,128], the authors of [117] proposed an alternating direction method of multipliers (ADMM)-based optimal power control scheme for an underlay uplink-cognitive HSTN. Both interference power and interference outage constraints were considered in real- and non-real-time applications.

By using game-theory-based maximization of throughput for global networks, the power control problem of a cognitive HSTN was studied. A distributed power control algorithm based on perfect channel estimation was proposed in [118] to obtain the Nash equilibrium (NE) of the game while ensuring the SINR requirements of the satellite links.

The authors of [119] proposed a short-term and a long-term optimal power control scheme to maximize the delay-limited capacity and minimize the outage probability for a cognitive LEO constellation with a terrestrial network by taking LEO satellite mobility into consideration. Similarly, an energy-efficient optimal power control scheme for an underlay LEO constellation-based cognitive HSTN with outdated CSI was presented in [120].

Introducing an effective energy efficiency metric employing delay QoS exponent for a cognitive HSTN with instantaneous CSI of a secondary transmitter and primary receiver, ref. [122] proposed a power allocation scheme by taking both the energy efficiency requirements and delay QoS provisioning into account.

A sensing-based power allocation scheme for an integrated wireless sensor network-cognitive HSTN was introduced in [108]. Perfect and imperfect sensing scenarios for seamless channel access of the cognitive satellite’s secondary user were considered.

### 3.2. Key Performance Parameters

The ergodic capacity of a multiple-antenna cognitive HSTN was analyzed in [143], where both the secondary terrestrial and primary satellite system had interference power constraints. The Meijer-G function-based analytical expression for the ergodic capacity of the secondary network was derived. The ergodic capacity and OP of an underlay-cognitive HSTN with secondary satellite and primary terrestrial network were also investigated in [132] by analyzing the impact of imperfect CSI with both proportional and peak interference constraints.

The outage performance of a cognitive HSTN with an interference constraint was investigated in [129] by using the Meijer-G function, and a novel closed-form expression of OP for the considered cognitive network was derived. The asymptotic results at a high SNR were also presented for both the proportional interference constraint and the peak interference constraint, revealing the important performance advantages, the achievable diversity order, and the coding gain of the cognitive HSTN.

In [130], the authors investigated the performance of a cognitive fixed satellite services (FSS) and mmWave terrestrial network to provide a general framework for the coexistence of an FSS and terrestrial network based on ITU recommendations and state-of-the-art channel models. Closed-form analytical OP expressions were also derived to understand the impact of various network parameters.

The outage performance of a multi-user spectrum-sharing HSTN was investigated and opportunistic scheduling of multiple users was proposed in [144]. Both direct and relay links from a primary satellite source to multiple terrestrial users and a secondary transmitter–receiver pair on the ground were considered.

To maximize the achievable rate of the terrestrial user by jointly optimizing the base station, UAV transmit power and UAV trajectory subject to the interference temperature threshold imposed on satellite networks and the UAV’s mobility constraints, ref. [145] presented a coordinated multi-point transmission (CoMP) architecture in a cognitive HSTN associated with UAVs.

The outage analysis of an overlay-cognitive HSTN in which an opportunistically selected secondary IoT network enables primary satellite–terrestrial communications in the presence of terrestrial and extraterrestrial combined interference was presented in [102].

The comprehensive outage performance of an overlay-cognitive multi-user satellite–terrestrial network was presented in [134], in which a primary satellite communicates with a selected terrestrial receiver via a secondary IoT network in the presence of interference from extraterrestrial sources (ETSs) and terrestrial sources (TSs).

Considering practical channel assumptions, including small-scale fading, the effect of interference due to spectrum sharing between the terrestrial and satellite systems in a cognitive HSTN was discussed in [109].

### 3.3. HI and Interference

The authors of [135] studied the performance limitations of a cognitive HSTN due to HIs and channel estimation error (CEE). Asymptotic analysis of OP and throughput at high SNRs was given. Similarly, the performance of a cognitive satellite–terrestrial cooperative network with HIs and outdated CSI in the presence of multiple primary users was investigated in [133], and the impact of practical HIs on a cognitive HSTN with multiple primary users was studied in [131].

In [124], for the HIs at the terrestrial users, an adaptive relaying protocol was proposed for both AF- and DF-based overlay-cognitive HSTNs. In [136], the impact of practical hardware impairments on an overlay-cognitive satellite–terrestrial relay network was investigated.

The authors of [125] introduced an AF-based cognitive satellite–terrestrial cooperative network where the terrestrial network was regarded as the primary system and the satellite network operated as the secondary system. The impacts of various system parameters on the SINR, SER, and OP performances of the system were also studied by restricting the transmit power of satellite communications with interference power constraints and applying MRC at the destination.

Finding the optimal transmit power under the interference constraint imposed by the satellite communications and the energy efficiency (EE)–spectral efficiency (SE) trade-off and power allocation associated with a cognitive satellite–vehicular 5G network, the authors of [114] proposed a unified EE–SE trade-off metric with a preference factor through which the priority level of EE/SE can be flexibly changed to adapt to dynamic surroundings.

Based on the similar EE–SE trade-off metric developed in [114], a power allocation scheme from the EE–SE trade-off perspective was formulated in [121] as an optimization problem that minimizes the utility function of vehicular communications while guaranteeing the interference power constraint imposed by satellite communications.

For dynamic power allocation based on instantaneous channel conditions, ref. [126] proposed a non-orthogonal multiple access (NOMA)-based cooperative overlay spectrum-sharing scheme in an HSTN for better spectrum utilization and tackling of the fairness issue using DF relaying protocol for the secondary terrestrial relay network.

The authors of [110] introduced a Vickery auction-based secondary relay selection for cooperative overlay spectrum-sharing satellite–terrestrial sensor networks with potential relays, e.g., 5G base stations, device-to-device nodes, and other ad-hoc networks. A comparative analysis of proposed relay selection schemes with traditional algorithms, such as maximum satellite–relay and relay–destination links, was provided. The same Vickery auction-based secondary relay selection scheme was investigated for an overlay-cognitive HSTN with NOMA for increasing capacity in [146].

The authors of [147] introduced a cost-effective, flexible, and high-throughput architecture of a cognitive satellite–air–terrestrial integrated network (SATIN) for IoT. Proposing a cooperative beamforming scheme to enhance the security and energy efficiency of IoT communications in cognitive SATINs, the main challenges to efficient resource management in terms of 3D spatial–temporal deployment of UAVs, interference management, and physical layer security were analyzed.

The performance for the coexistence of broadband satellite systems and terrestrial networks operating at the mmWave frequency was investigated in [127]. A constrained optimization problem was formulated by considering the availability of statistical CSI. Focusing on maximizing energy efficiency, an integrated cognitive HSTN with a distributed cooperative spectrum-sensing network was presented in [113].

In this section, we have reviewed key performance indicators and spectrum sharing in cognitive HSTNs. Performance-limiting factors such as HIs and interference and their mitigation techniques were studied. Cognitive HSTNs have been proposed due to their low cost, advanced network topology, and low complexity. However, cognitive HSTNs need continuous improvement in security performance, as there are more chances for attackers in cognitive radio technology. The data may be eavesdropped or altered without notice, and channels may be jammed or overused by adversaries. Major contributions to cognitive HSTNs are summarized in Table 5.

## 4. HSTN Applications

In this section, we discuss some of the typical applications of HSTNs, which may be exclusive to HSTNs due to incapabilities or limitations of conventional networks. Considering the capability of expanding network coverage, ensuring service continuity, and increasing network reliability, the following applications of HSTNs are discussed.

### 4.1. Emergency Communications

In emergency situations, communication support is essential for both public safety and disaster relief. Terrestrial communication may be paralyzed or destroyed in emergencies. In such desperate situations, terrestrial connectivity may be rendered useless or may even be obliterated. The 2011 Japan earthquake resulted in almost complete infrastructure disaster. VSATs were used to established communication links. A VSAT is a small earth station that receives and transmits real-time data via satellite [148]. Portable VSATs for disaster-resilient wireless networks [149] and software-defined radio-based multi-mode low-power and emergency-mode-adaptive bandwidth control VSATs [150] were developed to establish communication links in disaster-hit areas.

Although there are several ways to improve the stability and reliability of terrestrial networks, doing so requires a significant amount of time and resources. HSTNs are a promising solution to guarantee communication and can be utilized to establish emergency communication links and to construct a temporary WLAN to provide services to users through temporary BSs established through aerial platforms such as UAVs, balloons, or emergency ground vehicles.

### 4.2. Rural Coverage

One of the most important and promising applications of HSTNs is its expansion of coverage to rural, mountainous, and tough terrains. Based on their wide coverage and connectivity, HSTNs can provide services to previously unserved areas and populations. For areas with sparse users, mobile users can access the HSTN by their own terminals. Additionally, for residences or buildings in rural areas, fixed satellite antennas can be deployed to provide relatively broadband service for users inside based on 5G or Wi-Fi technologies.

Users can access the HSTN by BSs or other access points with satellite backhauls. Furthermore, the HSTN can also provide additional links to enhance the transmission in rural areas without broadband terrestrial networks.

### 4.3. Maritime Communication

Although humans mainly live on land, the sea is also of great importance for transportation, marine resource exploitation, and tourism. Traditional terrestrial networks can only enable communication offshore, but satellite systems have the wide coverage capability to span the entire sea region. By extending traditional terrestrial-based services to the sea area for different consumers, the HSTN can link the remote sea area with the land. Users onboard a cruise ship can access the same terrestrial services as those available on the ground thanks to a HSTN.

Additionally, gathering maritime data and maintaining maritime surveillance are crucial to ensure the safety and security of the maritime environment. An HSTN can provide efficient storage, transmission, and calculation of collected maritime information, improving the ability of continuous situational awareness of the sea [37].

### 4.4. Aerial Communication

An airborne network typically consists of balloons, airplanes, and/or UAVs, with aircraft having a critical need for broadband access to support passengers’ urgent communication requirements. However, current in-flight communication, such as Wi-Fi services, is rather capacity-limited, failing to satisfy the broadband communication demands of passengers.

Because of its broad coverage, an HSTN can provide continuous broadband service to passengers throughout flights, connecting the air with the ground. GEO or MEO satellites can be used to minimize frequent satellite handovers, while LEO satellites can be used to decrease communication latency. Additionally, the HSTN can be used to communicate with balloons and UAVs, which can be used for regional coverage, environmental monitoring, and border surveillance.

Some possible HSTN application cases were explored in this section. We have given a general overview of how HSTNs can expand coverage on land, in the air, and at sea. The importance of HSTNs in enhancing maritime coverage was outlined, and their potential for providing coverage over difficult terrain and sparsely populated areas was highlighted.

## 5. Key Cooperative and Cognitive Technologies

### 5.1. NOMA

NOMA has gained significance recently as an efficient access technique with the capability of improving spectral efficiency. NOMA is an essential enabling technology to meet the heterogeneous demands for low latency, high reliability, massive connectivity, improved fairness, and high throughput. Considering its capabilities to achieve better spectral efficiency and massive connectivity [151,152], the role of NOMA has been discussed in cooperative HSTNs.

#### 5.1.1. NOMA-Based Cooperative HSTN Networks

For integrated satellite–terrestrial networks in which the base stations and the satellite cooperatively provide services for ground users, the SINR and OP performances of a NOMA-based AF-relay HSTN were discussed in [153], and OP and ergodic capacity of a novel accessing technique known as cooperative NOMA were studied in [154]. By adopting pilot-based channel estimation, the impact of imperfect CSI was analyzed for a NOMA-based DF relay HSTN in [155]. A similar outage analysis and optimization of a NOMA-based HSTN with an AF relay under imperfect CSI was also studied in [156].

Using an opportunistic relay selection scheme, the impact of hardware impairment was studied in [157] for a two-way multiple relay and in [158] for DF relay-based HSTN. A similar impact study of HIs and partial rela6y selection on a NOMA-based integrated satellite–terrestrial multi-relay network was provided in [159].

The impacts of channel estimation error and channel fading parameters on the outage performance of a multi-user NOMA-based relay HSTN for both AF and DF relaying protocols were studied and compared with orthogonal multiple access (OMA) in [160]. To optimize the system capacity and reduce the complexity, the authors of [161] proposed a joint relay-and-antenna selection algorithm based on MIMO–NOMA for a DF relay-based HSTN.

To improve the performance of integrated satellite–terrestrial NOMA networks comprising multiple ground users with cooperative D2D communication, the authors of [162] proposed a pairing scheme for ground users by considering each user’s satellite channel and the terrestrial channels between users.

In [163], the authors investigated how well a novel kind of NOMA-based HSTN relay network performed during outages by taking into account uneven channel conditions to address potential difficulties in real-world scenarios where one user has a direct link to the satellite while the other does not and must instead rely on a DF multi-antenna relay to receive the satellite signal. A novel DF relaying protocol was also suggested to improve the performance of both users.

The authors of [164] presented a novel method for successfully integrating network coding (NC) and NOMA operations in a multi-user HSTN context. In contrast to other HSTN systems where the satellite is used for NOMA transmission, requiring CSI, the proposed system uses the satellite just for NC, negating the need for CSI and greatly simplifying the network. Since the satellite only needs the indices of the user-pairs and not their actual CSI to execute NC, the proposed approach does not incur additional CSI overhead. By doing this, improved performance is achieved while still being able to implement with little complexity.

In [165], the authors investigated the effectiveness of a cache-enabled HSTN aerial network, where the user could retrieve the required content files from the aerial nodes (ANs) or the satellite using the NOMA scheme. This was done to decrease transmission latency and facilitate the frequent updating of the files cached at the ANs. The outage probability and hit probability of the proposed network were specifically determined based on stochastic geometry, taking into account the uncertainty of the number and placement of ANs as well as the channel fading of users.

#### 5.1.2. NOMA-Based HSTN Cognitive Networks

A NOMA scheme to serve multiple secondary users simultaneously and improve the efficient use of limited resource in the secondary network was considered for a cognitive HSTN in [166] by analyzing the ergodic capacity to understand the performance limitations of the considered system.

Formulating a sum rate maximization problem, ref. [167] proposed NOMA-based beamforming in a cognitive HSTN operating at the Ka band, where a uniform planar array (UPA) was deployed at the base station (BS) to compensate for the path loss of high-frequency signals.

In [168], a NOMA-based power allocation scheme considering instantaneous channel conditions was proposed to improve the spectral efficiency of an overlay-cognitive HSTN. However, only a single PU with no direct satellite (DS) communication was considered.

Different from [168], the authors of [169] considered an overlay scheme that integrated the secondary terrestrial relay cooperation to with the primary DS communication in order to achieve a diversity gain for the primary network for a NOMA-assisted multi-user overlay-cognitive HSTN.

The performance of a cooperative NOMA-assisted underlay-cognitive HSTN wherein a secondary terrestrial network shares the spectrum with the primary satellite network to communicate with intended receivers was investigated in [170].

To mitigate the effects of inter-carrier interference (ICI) and intra-group interference (IGI) caused by bandwidth compression and power-domain multiplexing in a NOMA-based HSTN, an iterative and successive interference cancellation (ISIC) approach was proposed in [171] with symmetrical coding (SC) to avoid error propagation due to ISIC.

A user-pairing scheme to group multiple users into clusters and a joint optimization design of beamforming and power allocation in a downlink NOMA-based integrated satellite–terrestrial network operating at mmWave band was proposed in [172].

### 5.2. Beamforming

Beamforming is a signal processing technique used in antenna arrays to have spatial discrimination and filtering abilities [173]. However, beamforming does have limitations, such as high cost and computing resources. Multi-antenna beamforming is an effective means to mitigate co-channel interference and has been widely used in traditional fixed-spectrum-based wireless systems [174].

Beamforming techniques for hybrid satellite–terrestrial networks have been discussed in [175] to solve the problems of beamforming- and combining-based AF relaying in cooperative HSTNs.

Aiming to minimize the transmit power without compromising QoS, the authors of [176] investigated relay beamforming schemes for a cooperative HSTN, where a cognitive base station (CBS) with multi-antennas was used not only as a relay to assist with signal transmission between a satellite and a terrestrial primary user (PU), but also as a base station to transmit signals to multiple secondary users (SUs) with known CSIs for both uplink and downlink.

The outage analysis of a multi-antenna DF relay-based HSTN with beamforming was discussed in [177]. By applying single-antenna relays, the authors of [178] investigated cooperative beamforming under interference for an HSTN for wjocj a satellite transmits signals to a terrestrial mobile terminal (MT) through the relays.

Robust multi-objective beamforming was proposed in [179] for an integrated satellite and high-altitude platform network with imperfect CSI by formulating a sum rate maximization and total transmit power minimization trade-off.

For a spectrum-sharing HSTN in [180], an optimal beamforming scheme was designed by considering the impact of a non-linear power amplifier and imperfect CSI at the transmitter. An overview of the energy-efficient transmission, total power constraint, and per-antenna power constraint beamforming schemes for a UAV-relay-based HSTN were studied in [181].

A robust downlink hybrid beamforming scheme for a satellite–terrestrial integrated network was considered in [182] under the realistic assumption that the angular information of eavesdroppers was not perfectly known.

### 5.3. Software-Defined HSTNs

Software-defined radio (SDR) is a paradigm that promises to change the estate of the affair by breaking vertical integration, separating the network’s control logic from the underlying routers and switches, promoting (logical) centralization of network control, and introducing the ability to program the network [183]. SDR has the potential to overcome several HSTN challenges, such as unstable inter-satellite links due to fast movement of satellites, under-utilization of scarce spectrum, and lack of interoperability caused by the heterogeneities of hardware and software.

Dynamic configuration of satellite services can be performed by softwareization and virtualization as well as in-orbit adjustment of satellite coverage, operation frequency, bandwidth, and transmission power. For instance, in cases when satellites are required for electronic reconnaissance, the payload of radar will be scheduled, and related tasks such as signal processing, high-resolution imaging, and moving-target detection would be reconstructed using shared resources. The payload of satellites can be converted to a radio base station to give wireless connectivity in different circumstances when communication services are required. Additionally, the satellite software can develop autonomously, allowing for cutting-edge design and function changes without modifying the hardware. To adapt to the diverse networking requirements, SDR-enabled satellites with a universal hardware platform are able to control the parameters of the satellite radio signal, such as modulation mode, bandwidth, beamforming, frequency, and power, and can reconstruct functions deployed on the satellite, such as Access and Mobility Management Function (AMF), Session Management Function (SMF), and User Plane Function (UPF). SDR-based HSTNs can further improve communication on the move, IoT in remote areas, and communication among non-terrestrial platforms [184].

In [185], the SDR framework for integrated satellite terrestrial networks was investigated, and the agility of the SDR architecture was analyzed based on a three-layer satellite constellation. An SDR-based integrated satellite–terrestrial network was applied to vehicular networks in [186]. By performing network slicing for different segments, service continuality and QoS requirements for vehicles with high mobility were guaranteed. In [187], the traffic offloading problem for the integrated satellite–terrestrial network was investigated by relying on the SDR architecture. A detailed offloading scheme was proposed with extra offloading functions added to the control plane, which were specialized for offloading decisions, execution, and monitoring. Then, with the SDR control architecture, an end-to-end routing scheme was proposed for unified routing in the integrated satellite–terrestrial network [188], which can significantly reduce network congestion. In addition, due to the long propagation delay of satellite links, determining the location of controllers is more important and complex in the integrated satellite–terrestrial network. In [189], a joint controller and gateway-placing scheme was proposed to enhance reliability and reduce latency, which was validated with real topologies.

### 5.4. Physical-Layer Security

As a complex communication infrastructure comprising satellites in different orbits, UAVs, and other low-and high-altitude aerial platforms and terrestrial networks, an HSTN may have security risks and eavesdropping threats. Thus, methods of confidentiality shall be employed to prevent unauthorized data disclosure, as all the interconnected devices could suffer from this vulnerability [34].

The secure transmission of sensitive data via HSTNs continues to be a severe problem due to the broadcast nature of wireless communications. Secure communication has historically been ensured by cryptographic algorithms at higher tiers of the protocol stack based on the now-debunked premise that eavesdroppers have limited computational capabilities. Communications can be safeguarded at the physical layer by making use of wireless channel characteristics and cutting-edge signal processing techniques, which has generated a lot of interest in the field recently [190].

#### 5.4.1. Physical-Layer Security of HSTN Cooperative Networks

The secrecy capacity of a multi-antenna AF relay-based HSTN was investigated in [191] in the presence of a terrestrial eavesdropper by adopting a two-stage beamforming scheme, where the relay first combines the received signals from satellites with MRC and then forwards them to the destination and eavesdropper with zero-forcing processing. In the case of multiple relays, optimal and partial relay selection schemes were proposed in [192] to minimize the secrecy outage probability.

Using optimal relay selection to improve transmission security, the authors of [193] explored the physical-layer security of a DF-based multi-relay HSTN. In [194], maximum ratio combining and zero-forcing beamforming were used to ensure secrecy from multiple eavesdroppers in HSTN relay networks.

Considering a situation where the eavesdropping user can eavesdrop on both the satellite and the relay, the secrecy performance of an HSTN with multiple eavesdroppers was investigated in [195] with two selection schemes.

Considering a generalized and complicated setup of an AF relay-based downlink multi-user HSTN with a multi-antenna satellite and multiple unauthorized eavesdroppers, opportunistic scheduling of terrestrial users was used for non-colluding (N-COL) and colluding (COL) eavesdropper scenarios The ergodic secrecy capacity (ESC) performance of the considered system was investigated for the first time in [196].

Since zero-forcing and maximum ratio transmission beamforming were proposed in [194], which required instantaneous channel information of the downlink, the authors of [197] proposed a Rayleigh quotient-based beamforming scheme for secure transmission in a DF relay HSTN. Illustrating practical insights on the achievable diversity order of the system, ref. [198] investigated the secrecy performance of a downlink multi-user AF HSTN with opportunistic scheduling of terrestrial users.

The impact of a joint relay selection and user scheduling scheme for security against wiretapping attacks with multiple terrestrial relays, multiple terrestrial users, and multiple illegitimate eavesdroppers was studied in [199] for cooperative and single best eavesdropping scenarios.

Utilizing single and multi-relay selection schemes and only legitimate channel CSIs, the outage performance of an HSTN DF relay network where a satellite connects to a terrestrial destination through multiple relays at the appearance of an eavesdropper was studied in [200].

Reference [201] analyzed the physical-layer security of a downlink multi-user multi-relay HSTRN with a multi-antenna satellite under AF and DF relaying protocols. It used the best user–relay pair selection criteria for minimizing the secrecy outage probability (SOP) of the considered system for collaborative and non-collaborative eavesdropping scenarios. The work in [202] investigated the impact of an opportunistic user-scheduling scheme on the SOP for an integrated satellite–terrestrial relay network, where multiple terrestrial users and terrestrial eavesdroppers were considered in the system.

Addressing the vulnerability of an HSTN against eavesdroppers, a three-level AF/DF selection scheme (TLSS) was proposed in [203] for secure communication by applying null-space beamforming (NSBF) to the AF relay protocol and weighted fractional data carrying artificial noise (WFDCAN) to the DF relay protocol.

If the satellite is equipped with a single antenna, the eavesdropper in the service region is able to directly intercept the satellite transmission, resulting in a large eavesdropping rate. Thankfully, cooperative relays can cooperatively jam the eavesdropper. In [204], the optimal design of the jamming signals sent by relays to maximize the achievable secrecy rate for a cooperative HSTN under total relay power constraints was investigated.

A hybrid satellite–free space optics (FSO) cooperative system can find application in a number of scenarios due to its advantages in terms of fast and affordable deployment, wide coverage, and high throughput. Motivated by the latest advances in physical-layer security and the potential of the hybrid satellite–FSO system in various scenarios, the secrecy performance of satellite–FSO cooperative systems for cases of both AF and DF relay were investigated in [205].

Addressing hardware impairment on the secrecy performance of HSTN relay networks, the authors of [206] proposed an optimal relay-selection (ORS) scheme and analyzed its secrecy outage probability in the presence of hardware impairment along with traditional round-robin scheduling (RRS) as a benchmark scheme.

Most works based on the physical-layer security of UAV networks have considered the optimization problem. Very few works have analyzed the information-theoretic secrecy performance of a downlink point-to-point UAV network (e.g., [207,208]) without mobile relaying.

In [209], the authors introduced novel 3D-secure mobile UAV relaying for HSTN in the presence of an eavesdropper. By considering three opportunistic UAV relay-selection strategies, namely closest UAV relay selection, uniform UAV relay selection, and maximum signal-to-noise ratio UAV relay selection strategies, the secrecy performance of the system, probability of non-zero security capacity (PNZSC), and secrecy outage probability (SOP) were analyzed.

A beamforming scheme for maximizing the achievable secrecy rate (ASR) for a DF-based UAV relay equipped with multiple antennas was proposed in [210] to investigate secure transmission on an HSTN with multiple eavesdroppers.

Secrecy energy efficiency (SEE) is a key performance metric for the evaluation of security and energy efficiency in networks. To address rising energy costs and security requirements of communication networks, the authors of [190] looked into the problem of SEE optimization in satellite–terrestrial integrated networks by proposing novel hybrid analog–digital beamforming.

#### 5.4.2. Physical-Layer Security of Cognitive HSTN

As the computational ability of eavesdroppers increases, cryptography-based information security is threatened. Therefore, advanced signal-processing and communication techniques are required to protect the communication link from wiretapping. In order to realize secure communications with optimized resources, physical-layer security has been studied in various communication channels, such as multiple-input–single-output channels [211], multiple-input–multiple-output channels [212], relay networks, and cognitive channels [213].

Defining a physical-layer security framework for a cognitive HSTN where the terrestrial network functions as the secondary network and shares the downlink spectral resource with the satellite network, which functions as the primary network, the authors of [214] proposed the use of co-channel interference as a useful resource to enhance the secrecy performance of the satellite network.

Extending their system model in [214] to a more general case with multiple eavesdroppers, robust beamforming schemes to improve the physical-layer security (PLS) of a cognitive satellite terrestrial network and 5G cellular network coexisting with a satellite network operating at millimeter wave (mmWave) frequencies were proposed in [215,216], respectively. To maximize the secrecy rate of the primary user, a joint beamforming design at the satellite and the terrestrial base station for secure communication in a cognitive HSTN was proposed in [217].

For a cognitive HSTN in [218], a robust beamforming scheme was proposed to minimize the transmit power while satisfying constraints on the signal-to-interference-plus-noise ratio (SINR) at the satellite user and the terrestrial user, the leakage SINR at the eavesdropper, and the interference power recorded at the satellite user.

The work in [219] proposed a beamforming scheme for a cognitive satellite terrestrial network, where the base station (BS) and a cooperative terminal (CT) were exploited as green interference resources to enhance system security performance in the presence of unknown eavesdroppers. Different from [217,218], only imperfect channel information of the mobile user (MU) and earth station (ES) was assumed.

Focusing on resource allocation and utilizing interference from the terrestrial network as a green resource to minimize the transmit power while satisfy the secrecy rate constraint of the primary user, the information rate constraint of the secondary user (SU), and the total transmit power constraint, ref. [220] investigated secure transmission in a cognitive HSTN.

Extending [216,217,218] to a more practical and general scenario with multiple eavesdroppers and with multi-cell terrestrial networks (multiple users per cell) coexisting with a satellite network, ref. [221] studied secure beamforming and artificial noise (AN) techniques for a cognitive satellite–terrestrial network, where a satellite system termed as the primary network under the interception of multiple unauthorized eavesdroppers shares its spectrum of resources with multi-cell secondary terrestrial networks.

For a software-defined architecture and cognitive HSTN operating in mmWave frequencies, joint satellite/base station beamforming schemes were proposed in [222]. The proposed beamforming schemes exploit the interference from the secondary network as a green source to enhance the performance of physical-layer security for the primary network without deteriorating the performance of the secondary network.

By adopting artificial noise (AN) and applying the weighted Tchebycheff approach to formulate a multi-object optimization problem, ref. [223] proposed a beamforming scheme based on multi-objective optimization (MOO) for a cognitive HSTN.

The secrecy energy efficiency maximization problem in an HSTN was investigated in [224]. To maximize the achievable secure energy efficiency of the Earth station under imperfect wiretap CSI while satisfying a secrecy constraint on the Earth station, data rate requirements for the cellular users, and power budgets of the satellite and base station, a secure beamforming scheme for rate-splitting multiple access (RSMA)-based cognitive satellite terrestrial networks in the presence of multiple eavesdroppers was presented in [225].

Using a cooperative eavesdropping scheme, secure performance of a wiretap satellite–terrestrial network with multi-eavesdroppers was investigated in [226].

Unlike the fully digital transmission beamforming presented in [214,216], the authors of [227] analyzed the physical-layer security of a fifth-generation (5G) cellular system operating at millimeter wave (mmWave) frequency in a satellite–terrestrial network by employing a hybrid analog–digital precoder at both the base station (BS) and a satellite-proposed-penalty dual-decomposition (PDD)-based secure beamforming algorithm.

The security problem in cognitive satellite–terrestrial wireless networks was studied in [228], and a power-efficient secure beamforming algorithm for improving the secrecy rate of the system was proposed. The hybrid network included the satellite communication network and the terrestrial secondary network.

With the capability of converting the received RF signals into electricity, there has been an upsurge of research interests in wireless information and power transfer (WIPT) techniques. Exploring a joint satellite–base station beamforming design for integrated satellite–terrestrial networks, a secure WIPT in an integrated satellite–terrestrial network operating at high frequency was investigated in [229].

A robust secure beamforming and power splitting scheme for mmWave cognitive HSTNs with simultaneous WIPT under an angle-based CSI error model were presented in [230] to solve a weighted sum maximization problem except for the max–min fairness problem.

Extending [226], a practical secrecy model was designed in [231] for land-mobile satellite (LMS) communication networks with multiple legitimate users and multiple eavesdroppers. The authors proposed a cooperating scheme for legitimate users to receive the main signal for colluding and non-colluding eavesdropping scenes.

Different from [218,220], the authors of [232] considered a joint precoding design at the satellite and base station for secure multicast transmission in cognitive satellite–terrestrial networks with multiple multi-antenna eavesdroppers, where the satellite (SAT) delivers a common confidential message to a group of legitimate users, and the interference from the terrestrial base station (BS) was utilized to strengthen the security of the satellite link.

For a spectrum-sharing-aided HSTN that can utilize the shared spectrum in the presence of an eavesdropper, a specifically designed satellite scheduling scheme was proposed, and the security–reliability trade-off (SRT) of the mobile satellite services (MSS) scheme was analyzed in [233] to guarantee wireless transmission of the satellite system against eavesdropping attacks in the face of co-channel interference generated by the terrestrial communication system.

To maximize the secure EE for the primary satellite network while satisfying the allowable signal-to-interference-plus-noise ratio requirements of the secondary and primary users and remaining within the transmit power limitation of both the satellite and the terrestrial base station, ref. [234] investigated the secure energy efficiency optimization problem in a cognitive satellite–terrestrial network with a capable eavesdropper.

In [235], the secrecy outage performance of multi-antenna satellite–terrestrial transmissions was investigated. The closed-form expressions for the exact and asymptotic SOP were derived while considering the randomness of the positions of legitimate and eavesdropping receivers.

In this section, we have provided an overview of the key techniques of NOMA, beamforming, software-defined radio, and physical-layer security and their utility in HSTNs. The role of NOMA as an efficient access technique capable of improving spectral efficiency was discussed, as was the mitigation of co-channel interference through the use of multi-antenna beamforming and the use of software-defined radio in HSTNs for centralization and network control. Keeping in mind security risks and eavesdropping threats due to complex communication infrastructure, we have also provided an overview of the physical-layer security of HSTNs in this section.

## 6. Open Issues

In this section we identify several key challenges and open issues from the aspects of network architecture, performance enhancement, and optimization.

### 6.1. Multi-Layered Networks

GEOs, MEOs, and LEOs operate in distinct orbits, with high- and low-altitude aerial platforms, such as UAVs, airships, and balloons, operating at different levels, and a terrestrial network establishing another layer on the ground. Because of the various requirements of different segments in such a complex layered infrastructure, the overall system performance may be impacted. Maintaining necessary QoS standards needs extensive knowledge, which is a major problem in HSTNs.

### 6.2. Latency in Satellite Networks

Communication latency is a crucial and fundamental performance indicator for wireless communication that ensures service quality. The goal of communication network development, from 1G to 5G, has always been to cut down on communication latency. In general, delays in transmission, propagation, processing, and queuing make up communication latency. While the other three types of delays may be equivalent for satellite and terrestrial networks, satellite networks have substantially longer propagation delays than terrestrial networks because of their high-altitude orbits.

In satellites, there are two types of latency: fixed, owing to propagation delay; and dynamic, due to transmission, processing, and queuing delays [31]. Due to their high altitude and mobility, the inherent latency of GEO and non-GEO satellites may impact overall system performance. Furthermore, each link in the infrastructure may have its own processing time, causing additional processing delays. Designing an effective HSTN architecture that does not compromise system performance while also taking into account the satellite’s and associated systems’ inherent latency is a difficult task.

### 6.3. Channel Modeling and Measurement

Because of the long propagation distances and mobility of satellite and aerial networks and the mobility of terrestrial networks for mobile nodes, propagation channel modeling and channel measurement in HSTNs is a critical and challenging issue that requires careful attention due to its importance and the many different environment-related challenges that affect wireless propagation channels in HSTNs.

### 6.4. Security

Despite ongoing efforts to provide secure communication in HSTNs, the intrinsic nature of wireless communications makes it subject to assaults such as denial of service (DoS), spoofing, jamming, and other forms of interference. Physical-layer security (PLS) in HSTNs is an emerging framework in addition to standard cryptographic approaches. However, because of the sophisticated and complicated nature of today’s eavesdroppers, using the PLS architecture for secure transmission in HSTNs remains critical.

### 6.5. Link Conditions

Satellite links are hampered not just by the large propagation delay but also by the complicated link circumstances. The satellite communication link must pass through the atmosphere in order to reach the ground because satellites are propelled through space at speeds of hundreds to thousands of kilometers per hour. As a result, weather variables, including rain, clouds, water vapor, and fog, can affect satellite communication systems. Rain attenuation is the most important factor for spectrum over 10 GHz [51].

Different from terrestrial links, satellite links may experience severe channel fading due to poor weather, resulting in a breakdown in communication. Considering the long propagation delay of satellite links, obtaining timely and accurate channel state information is more difficult compared with terrestrial networks. MEO/LEO satellites travel at a rapid rate in relation to the earth. The satellit-e-ground linkages thus undergo a greater Doppler shift, a greater phase shift, and a faster time variation.

An HSTN relay network architecture can be used to improve transmission when the communication links between the satellite and users are unstable due to unfavorable weather circumstances or high channel dynamics because of the complex characteristics of satellite link conditions. Nevertheless, the link conditions are seriously unbalanced in a relay network for an HSTN, and this imbalance must be addressed, as the relaying performance is determined by the “weakest link”.

### 6.6. Path Selection and Routing

Unified routing protocols are necessary to allow integrated routing across various network components for consistency and efficiency. Conventional IP protocols, however, cannot be easily implemented because they were created for static terrestrial networks. If the routing strategy is not adjusted in a timely manner and in accordance with the network’s time-varying topology, routing oscillation may result in frequent connection disruptions. Additionally, due to the satellite’s extensive coverage, the routing table in the integrated satellite–terrestrial network may be huge, which makes it more difficult to determine the best routing techniques. When choosing the routing paths for various types of traffic, it is necessary to take into consideration the various propagation delays and conditions of terrestrial and satellite links. Low-latency pathways are preferred in order to meet users’ QoS requirements.

### 6.7. Resource Management

The basic communication architecture of the terrestrial network and the satellite network differs, resulting in resource configurations that differ between the two networks. While spectrum and power resources are shared by both networks, orbit and gateway resources are unique to the satellite network due to its top-down nature. To meet the QoS requirements of various users and traffic, all distinct resources in HSTN must be managed collaboratively. If we implement resource allocation for each network separately or simply apply traditional resource management strategies to the integrated network, network performance will suffer greatly due to inefficient resource utilization. Given the differences in resource characteristics between satellite and terrestrial networks, novel resource management mechanisms must be developed.

### 6.8. Handovers and Mobility Management

When users move out of the coverage area of the original connection, a handover is used to transfer the connection between cells. Handovers occur more frequently in an HSTN due to the orbital motion of MEO and LEO satellites and the mobility of mobile users. Thus, efficient mobility and handover management is critical to ensure service continuity and achievement of QoS requirements. A standard handover procedures in each network can be used to execute handover when it only happens between terrestrial or satellite networks. However, changeover across various networks is complex and more challenging.

In cooperative HSTNs, where the satellite network and the terrestrial network work together to deliver improved communication services for ground users, more complicated handovers may take place. Since users are serviced by two cooperative connections in such an architecture, the cooperation must be transferred to the new connection when conducting a handover for one of the connections. The complexity of handovers is increased by the possibility of simultaneous handovers of the two connections under more unique circumstances.

### 6.9. Traffic Offloading

In heterogeneous networks, data traffic can be offloaded from one network to another to prevent congestion. When the terrestrial network is insufficient or overloaded, cooperating HSTNs can also use the satellite to offload some of the data traffic. In comparison to traditional traffic-offloading techniques in terrestrial networks, satellite-based traffic offloading is more complex because of the particular characteristics of satellite networks.

Satellite–ground links are unreliable and the connection times are short due to the fast mobility of MEO and LEO satellites. Full data flow may be offloaded sequentially by various links at various times. As a result, to accommodate the various offloading links, dynamic offloading techniques are required. Due to satellites’ lengthy propagation delays, connection latency should be taken into account; this is in contrast to terrestrial offloading, where link capacity is the primary factor in reducing congestion.

In order to maximize network performance while maintaining user QoS, the offloading method must be devised based on the various offloading links because satellites in different orbits have various propagation delays. A satellite is not the sole option for traffic unloading in an HSTN, either. Offloading of traffic from both satellite and terrestrial sources can be combined to boost network performance even more.

In this section, we have provided a comprehensive overview of open challenges and issues that are important for the implementation, utilization, and performance of HSTNs. Various open issues have been discussed in the context of maximizing performance of the HSTNs and overcoming the inherent problems with different network segments.

## 7. Conclusions

This survey has provided a thorough examination of the current state-of-the-art in the field of HSTNs. We have presented various HSTN network architectures and provided an in-depth overview of current work in cooperative and cognitive HSTN network architectures. NOMA and beamforming have been reviewed. There is also an overview of the work done in physical*layer security of various HSTN network architectures. Various potential applications of HSTN have been presented. In the end, open issues have been discussed to conclude our work.

## Figures and Tables

**Figure 1 sensors-22-08544-f001:**
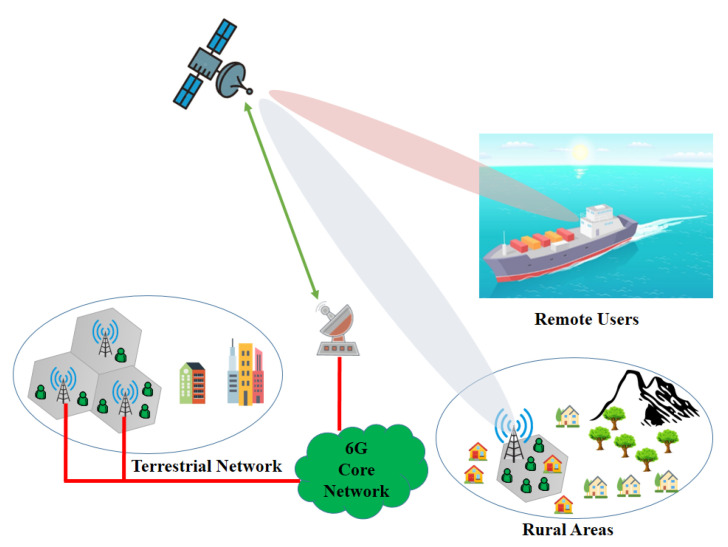
HSTN architecture.

**Figure 2 sensors-22-08544-f002:**
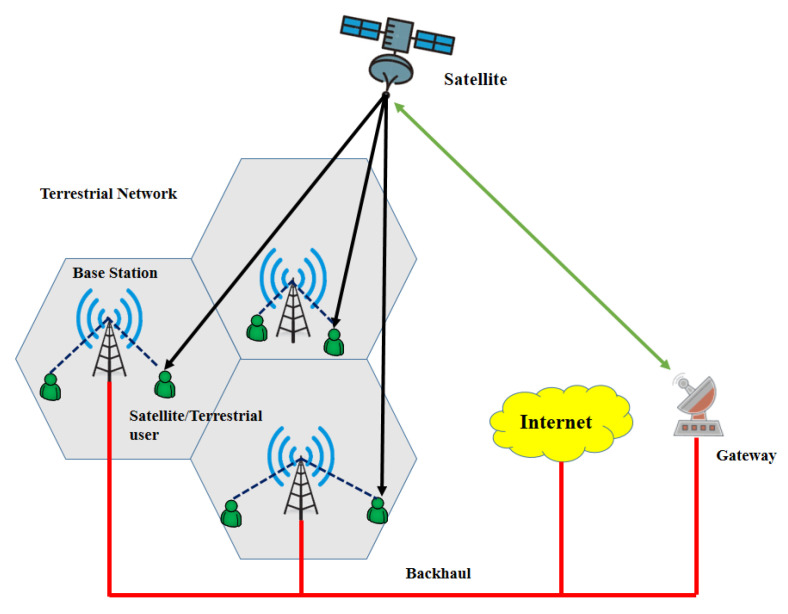
Coordinated HSTN.

**Figure 3 sensors-22-08544-f003:**
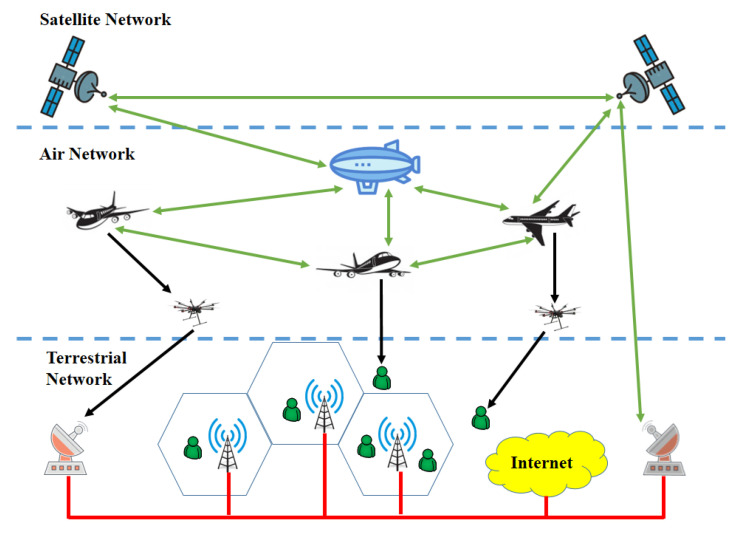
HSTN with aerial networks.

**Figure 4 sensors-22-08544-f004:**
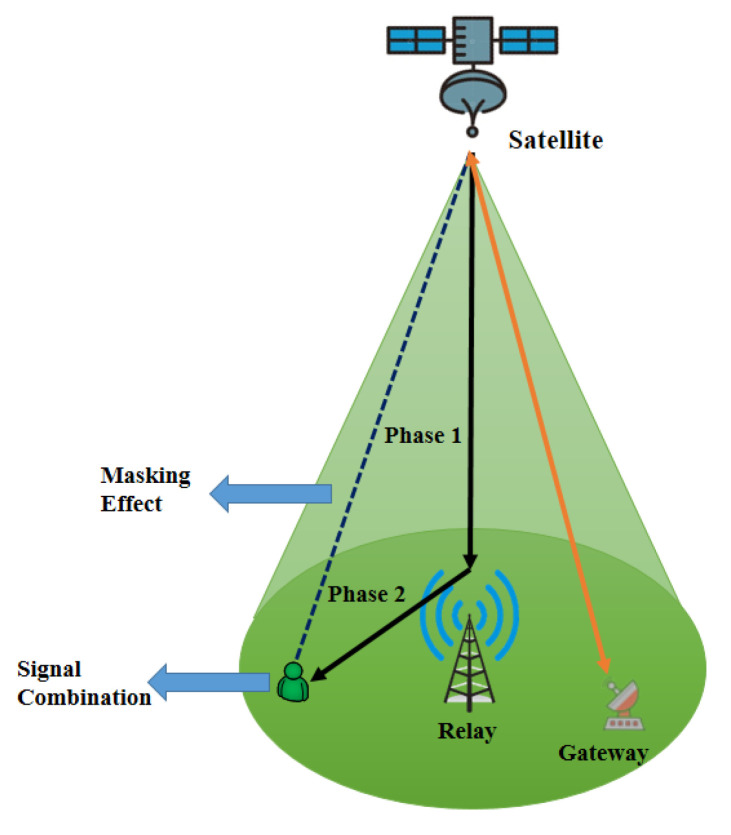
HSTN with relaying structure.

**Figure 5 sensors-22-08544-f005:**
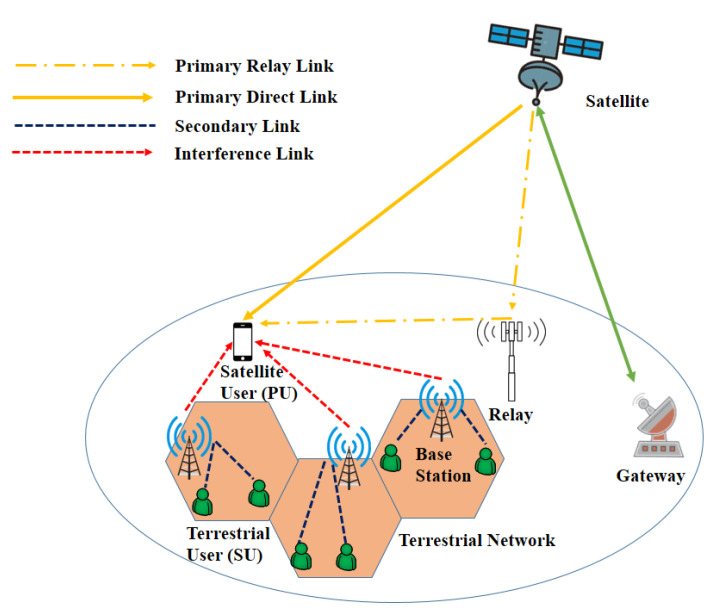
Cognitive HSTN.

**Table 1 sensors-22-08544-t001:** Comparison of satellite and terrestrial networks.

Network	Latency	Speed
(Satellite) Starlink	20 ms	80–150 Mbps
(Satellite) Oneweb	70 ms	7 Gbps per satellite
(Satellite) Konnect VHTS	600–700 ms	500 Gbps
(Terrestrial) 4G	20–30 ms	300 Mbps
(Terrestrial) 5G	<10 ms	10–50 Gbps

**Table 2 sensors-22-08544-t002:** Comparison of surveys of HSTNs.

Reference	Contributions	Year
[28]	Handover schemes in satellite networks	2006
[29]	Comparison of different mobile–satellite systems	2010
[30]	Design parameters, operational solutions, and inter-satellite communication of small satellite systems	2016
[31]	State-of-the-art architectures of converged satellite–terrestrial systems	2016
[32]	Optical communication techniques in integrated satellite–terrestrial network to mitigate atmospheric side effects	2016
[33]	State-of-the-art space–air–ground integrated networks	2018
[34]	Convergence of satellite–terrestrial networks, performance evaluation, and simulation platforms	2019
[35]	Non-terrestrial network (NTN) architectures and NTNs in cellular communications.	2020
[36]	Future research directions for a cell-free, hierarchical, decoupled HSTN and open issues to envision an agile, smart, and secure HSTN	2021
[37]	An environment-aware and service-driven hybrid satellite–terrestrial MCN for future maritime communications	2021
Our work	Comprehensive review of cooperative and cognitive HSTNs centered on key technologies with their performance and open issues	2022

**Table 3 sensors-22-08544-t003:** List of major contributions to coordinated HSTN networks.

Reference	System Model	Performance Metrics	Other Factors
[46]	Multi-antenna satellite communicating with a ground receiver via multiple AF 3D mobile UAV relays under opportunistic relay selection	OP comparison of static and mobile relays	Fixed altitude of UAVs
[47]	HSTN with a DF-based aerial relay and a group of terrestrial receivers	Approximate OP expressions, interference, and non-interference coverage analysis	Transmission power and Time average between satellite–relay and relay–destination links
[48]	Three-layer space–air–ground integrated network	Link delay, bandwidth, and connection time	Network traffic
[49]	DF-based terrestrial relay HSTN	MGF-based ASER analytical expressions at high SNR	CCI
[50]	Three-node AF terrestrial relay-based HSTN	ASER performance	Path loss, multipath fading, and single and multiple CCI at relay
[51]	Three-node distributed space–time coded AF–terrestrial relay-based HSTN	ASER performance	Fading
[52]	Three-node AF hybrid satellite–terrestrial cooperative system with estimated channel gains	ASER and diversity-order calculations	Unavailable CSI
[53]	DF–terrestrial relay-based HSTN	OP analysis	Rain attenuation
[39]	Channel characterization for space/air communications according to the 3GPP and ITU guidelines and related system parameters	Average capacity and OP	Service continuity, ubiquity, and scalability
[45]	Multicast transmission scenario in space–air–ground integrated network	Maximum sum rate of all users	QoS requirements
[42]	UAV integration with existing MCNs	Minimum ergodic achievable rate bits/s/Hz	Coordination issues and limited CSI
[43]	Hierarchical satellite–UAV–terrestrial MCN	Energy consumption	Slowly varying large-scale CSI
[44]	Hybrid maritime network consisting of mobile users (ships), UAVs, terrestrial BSs, and satellites	Minimum ergodic achievable rate bits/s/Hz	Available CSI

**Table 4 sensors-22-08544-t004:** List of major contributions to HSTN relay networks.

Reference	System Model	Performance Metrics	Other Factors
[58]	AF-based ground relay and no direct satellite–destination link	SER and SINR	Co-channel interference
[59]	AF-based terrestrial relay	SINR, OP, SNR	Heavy shadowing
[60]	AF terrestrial relay	SINR and BER	CCI and noise
[61]	AF relay-based multi-user HSTN	OP under high and low CCI	CCI
[62]	Multi-antenna multi-user HSTN relay network	OP, EC, and achievable diversity order	Outdated CSI and CCI
[63]	Fixed-gain multi-user AF relay network	EC	CCI
[64]	Two-hop downlink multi-user HSTN AF relay network	Average SEP	Outdated CSI and CCI
[76]	Fixed-gain AF relaying HSTN	ASER, EC, and SNR	Channel parameters
[78]	Multi-user multi-relay HSTN	OP	High SNR and best user–relay selection
[79]	Selective DF relay-based HSTN	Average SER	Satellite elevation angles
[80]	Single antenna multi-relay HSTN	EC	Available LOS between satellite and user
[81]	Multiple-relay HSTN with mobile destination	Ergodic channel capacity analysis	Impact of channel conditions and cooperative protocols
[82]	Selective DF multi-relay HSTN	OP	Known CSI and channel parameters
[83]	AF multi-relay HSTN with wireless caching	OP	No LOS between satellite–destination
[84]	Single-antenna multi-relay HSTN	OP	Multipath and shadowing
[85]	Multi-relay land mobile system	OP	Known CSI and correlated fading between relay and user
[86]	single-user multi-relay HSTN	OP and throughput	Hardware impairments
[87]	Two-way multi-relay multi-antenna HSTN	OP and throughput	Hardware impairment, number of antennas, and number of relays
[88]	DF-based multi-relay HSTN	OP	Hardware impairments
[89]	Uplink land-mobile satellite system with multiple relays	OP and throughput	Hardware impairment, interference, and CCI
[90]	Dual-hop mmWave multi-user AF multi-relay HSTN	OP	Rain attenuation
[91]	AF CDMA-based HSTN	OP	Increased number of relays and LOS link
[92]	Cooperative HSTN with multiple mobile relays	ASEP and OP	Known CSI and slow fading
[93]	Multi-user integrated satellite—UAV aerial relay	Outage analysis and OP	High SNR
[94]	Dual-hop satellite relay network	OP	Rain attenuation
[95]	Multi-antenna 3D UAV relay-based HSTN	OP	Additive white Gaussian noise (AWGN) at all receiving nodes
[96]	ARMA-based SAGIN	Link congestion	Network traffic
[97]	3D UAV relaying HSTN	OP	Receiving nodes corrupted by AWGN

**Table 5 sensors-22-08544-t005:** List of major contributions to cognitive HSTN networks.

Reference	System Model	Performance Metrics	Other Factors
[104]	Coexisting satellite and terrestrial communication system with distributed base stations	Average achievable rate	Limited information exchange between satellite and terrestrial network
[105]	Overlay-spectrum-sharing cognitive HSTN with primary satellite and secondary terrestrial network	OP analysis	Multiple secondary networks within same spectrum and SNR
[107]	EH-based overlay-cognitive HSTN spectrum-sharing system with primary satellite and secondary terrestrial network	OP, throughput, and energy efficiency	Heavy and average shadowing
[115]	Underlay-cognitive HSTN with primary terrestrial and secondary satellite network	OP and delay-limited capacity	Transmit power limits, interference power constraints, shadowing and fading
[112]	Uplink-underlay-cognitive HSTN consisting of one primary terrestrial network and one secondary satellite network	Energy efficiency of real-time and non-real-time applications	Average interference power and peak and average transmit power
[118]	Cognitive HSTN with FSS primary and terrestrial secondary network	Throughput and actual SNR	Perfect channel estimations and SINR
[119]	Underlay-cognitive LEO constellation HSTN	OP and delay-limited capacity	LEO satellite mobility, large free space loss, available CSI
[143]	Multiple-antenna cognitive HSTN with terrestrial secondary and satellite primary network	Ergodic capacity	Imperfect CSI, channel parameters, and antenna configuration
[130]	Cognitive broadband HSTN with primary FSS and secondary terrestrial network	OP analysis	Cross-channel interference from satellite
[145]	UAV-aided cognitive HSTN with primary satellite and secondary terrestrial network	Throughput and UAV power allocation	UAV mobility
[102]	IoT-enabled overlay-cognitive HSTN	OP analysis	Terrestrial and extraterrestrial interference
[135]	Cognitive HSTN relay network with multiple-user primary terrestrial network	OP, throughput, and signal-to-noise-plus-distortion-and-error ratio	HIs and channel estimation errors
[114]	Underlay-cognitive satellite–vehicular network	Energy efficiency and spectral efficiency	Interference constraint
[147]	Cognitive satellite–aerial–terrestrial network	Energy efficiency	UAV deployment, physical-layer security, interference management, cooperative beamforming, and non-cooperative beamforming
[127]	Cognitive broadband satellite system and mmWave cellular network	OP and EC	High SNR
[113]	Integrated sensing-based cognitive satellite terrestrial network	EE (bits/Hz/J)	Computational complexity

## Data Availability

Not applicable.

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
