# Peer review of "Hybrid Satellite–Terrestrial Networks toward 6G: Key Technologies and Open Issues"

_sensors, 2022, doi:10.3390/s22218544_

Round 1

Reviewer 1 Report

This paper provides a comprehensive survey on hybrid satellite-terrestrial networks. Overall the discussion is comprehensive and the manuscript is well-written. Some minor suggestions are given as follows:

1. Some abbreviations are used before mentioning the full name, e.g., QoS in line 23 Page 1. Some abbreviations have no full name, e.g., LEO, MEO and GEO in the whole manuscript. Similar problems should be checked and fixed.

2. The format of "36,000 km" in line 38 and "36 000 km" in line 42, page 2 should be united.

3. The launch progress of Starlink and OneWeb can be further updated in line 103-114 Page 3.

4. The idea of SDN has been seen as a promising solution for many challenges in satellite-terrestrial integrated networks. The authors may want to add a subsection for relevant discussion by referring to

Yuan S, Peng M, Sun Y, et al. Software defined intelligent satellite-terrestrial integrated networks: Insights and challenges[J]. Digital Communications and Networks, 2022.

Reviewer 2 Report

The authors collect in several tables all the possible references concerning all the possible aspects of hybrid satellite-terrestrial networks.  Note that in this collection a huge variety of the possible aspects and problems are put together, ranging from channel estimation to cooperative technologies, to security issues, etc. Just a huge bibliographic collection without any useful contribution. No need for another review, after more than ten already published (see Table 1) in the last years.

The term 6G is not clear, a clear definition of what is 6G should be provided, since there is no actual standardization process labelled with the 6G label, but more important the authors should specify which are the peculiarities of 6G that are so new that cannot be implemented in the last releases of 5G standards.

Reviewer 3 Report

REVIEW

The idea of the article is laudable and the subject is interesting. Despite its many shortcomings, I suggest that it be improved and, after correction by the Authors, it can be published.

1.     It is worth noting that the concept proposed by the Authors is not new. The Authors own contribution should be clearly indicated at the beginning of the article in addition to the main purpose of the paper. The article is a survey, so it should clearly say what the contribution is and what is not. Besides, the article should be supplemented with the classification on the basis of which comparisons were made. At the moment it mixes past and present technologies. The solutions presented by the Authors usually concern systems that are many years old (it is worth adding information on how old the systems included in them are). This would supplement the article. Only then there will be no doubt about its usefulness for science as a survey paper.

2.     The Authors write in very general terms in the context of the technologies mentioned and do not indicate the topology of satellite networks. It is worth recalling that satellite networks can operate in many topologies (mesh, point-to-point, star and hybrid).

3.     The Authors mention latency or attenuation, which is high compared to terrestrial links. The main conclusion is that it would be worthwhile to systematize the article by comparing all parameters (the same) for different networks. This requires improvement.

4.     For the most part, the Authors provide basic information, of which the level of knowledge is basic. Nevertheless, such articles that organize the state of art of are also needed. However, the literature review needs to be more complete, as the Authors omit many recent items (concerning important world projects). Hence the suggestion to fill in the gaps in the literature review of recent years, especially. Some examples of projects are included below.

5.     Certainly, to counteract the effects of various adverse events, including natural disasters, it is necessary to look for fast and reliable ways to ensure communication. The European Union forces operators to use backup lines, which is applicable in the case of unpredictable events such as floods or earthquakes. In that case, satellite systems are most often used. In the case of adverse events, including natural disasters, the use of such links is essential to ensure stability of the functioning of the state, and on a microscopic scale, for example, to ensure contact with evacuated people. It should be emphasized that this topic is so important that it was the subject of European Research Project COST IC0802 and the construction of Global Integrated Networks (including GMES & Disaster Management and Relief), in which many European countries participated.

See more: https://www.cost.eu/actions/IC0802/ [footnote worth adding to the list of references].

In practice, Japan can be used as an example of the need for emergency links and crisis management (earthquakes occur there). In 2011, the ground infrastructure was almost completely destroyed. At that time, satellite systems were used and their benefits were recognized. Satellite links are also used in locations that are difficult to access, such as the Tuvalu Islands (an island state), where in recent years VSAT-based systems have been implemented, which has significantly improved the level of medical care in the country. Another example is the use of this technology to provide global connectivity (the cheapest way). As examples can be:

Goncharenko, Yu.Yu., Goncharenko, D.G., Divizyniuk, М.М. (2012). On the problem of calculating the range of acoustic information from open areas. Scientific and technical collection “Legal, regulatory and metrological support of the information protection system in Ukraine”. Kyiv: State Service for Special Communications and Information Protection of Ukraine NTUU “KPI”, Issue. 1 (23). p. 29-35.

Azarenko, O., Honcharenko, Y., Divizinyuk, M., Mirnenko, V., Strilets, V., Wilk-Jakubowski, J. L. The influence of air environment properties on the solution of applied problems of capturing speech information in the open terrain. Journal of Scientific Papers «Social Development and Security», Vol. 12(2), pp. 64-77, 2022. https://doi.org/10.33445/sds.2022.12.2.6.

Y. Ma, H. Jiang, J. Li, H. Yu, C. Li and D. Zhang, “Design of Marine Satellite Communication System Based on VSAT Technique”, 2021 International Conference on Computer, Internet of Things and Control Engineering (CITCE), 2021, pp. 126-129, doi: 10.1109/CITCE54390.2021.00031.

Wilk-Jakubowski, J.Ł. Information Systems Engineering Using VSAT Networks. Yugoslav Journal of Operations Research, [S.l.], Vol. 31(3), pp. 409-428, 2020. ISSN 2334-6043. Available at: < http://yujor.fon.bg.ac.rs/index.php/yujor/article/view/833/785>. https://doi.org/ 10.2298/YJOR200215015W

Wilk-Jakubowski, J. Predicting Satellite System Signal Degradation due to Rain in the Frequency Range of 1 to 25 GHz. Polish Journal of Environmental Studies”, Vol. 27(1), pp. 391-396, 2018, ISSN: 1230-1485. https://doi.org/10.15244/pjoes/73906

Wilk-Jakubowski, J. Total Signal Degradation of Polish 26-50 GHz Satellite Systems Due to Rain. Polish Journal of Environmental Studies, Vol. 27(1), pp. 397-402, 2018, ISSN: 1230-1485. https://doi.org/10.15244/pjoes/75179

Sasanuma, M., Uchiyama, H., Nagoya, T., Furukawa, M., Motohisa, T. Research and development of very small aperture terminals (VSAT) that can be installed by easy operation during disasters – Issues and the solutions for implementing simple and easy installation of VSAT earth station. IEICE, 112 (440) (2013).

Seumatsu, N., Kameda, S., Oguma, H., Sasanuma, M., Eguchi, S., Kuroda, K. Multimode SDR VSAT against big disasters. European Microwave Conference (EuMC 2013), Nuremberg, Germany, 2013.

Eguchi, S., Kameda, S., Kuroda, K., Oguma, H., Sasanuma, M., Suematsu, N., Multi-mode portable VSAT for disaster-resilient wireless networks. Asia Pacific Microwave Conference (APMC 2014), Sendai, 2014.

There are many publications on this topic that present practical examples of such solutions (some examples are given above).

6.     The Authors have rightly pointed out many limitations in the technology. However, the economic factor was not presented. There is no data source (i.e., footnotes) so that the cost of building a global data transmission system can be estimated in tables and figures using the technologies mentioned.

7.     The Authors do not mention that there are satellites with a capacity of 90 Gbps and even 140 Gbps! The VHTS Konnect satellite is supposed to have 500 Gbps. In fact, due to frequency multiplexing, some geostationary satellites (HTS) parameters match the speeds offered in terrestrial links (50 Mbps for downlink).

The article also has editorial shortcomings (e.g., pages should not end with drawings).

After correction, the article may be published at the Editor's decision. I believe that due to the very important subject matter and the amount of work done by the Authors, it should be published.

Round 2

Reviewer 2 Report

The authors reply to the request of the reviewer in a sufficient way.